**Ocean Alkalinity Enhancement (OAE) does not cause cellular stress in a phytoplankton community of the sub-tropical Atlantic Ocean**

Librada Ramírez[1*], Leonardo J. Pozzo-Pirotta[1], Aja Trebec[2], Víctor Vázquez-Manzanares[1], José L. Díez[1], Javier Arístegui[2], Ulf Riebesell[3], Stephen D. Archer[4], María Segovia[1].

[1]Department of Ecology, Faculty of Sciences, University of Malaga, Málaga, Spain.

[2]Instituto de Oceanografía y Cambio Global (IOCAG), Universidad de Las Palmas de Gran Canaria (ULPGC), Las Palmas, Spain.

[3]GEOMAR Helmholtz Centre for Ocean Research Kiel, Kiel, Germany.

[4]Bigelow Laboratory for Ocean Sciences, East Boothbay, Maine, United States.

*Correspondance to*: Librada Ramírez (librada@uma.es)

**Abstract**

A natural plankton community from oligotrophic subtropical waters of the Atlantic near Gran Canaria, Spain was subjected to varying degrees of ocean alkalinity enhancement (OAE) to assess the potential physiological effects, in the context of the application of ocean carbon dioxide removal (CDR) techniques. We employed 8.3 m$^3$ mesocosms with a sediment trap attached to the bottom, creating a gradient in total alkalinity (TA). OAE was based on the addition of carbonates (NaHCO$_3$ and Na$_2$CO$_3$) The lowest point on this gradient was 2400 μmol · L$^{-1}$, which corresponded to the natural alkalinity of the environment, and the highest point was 4800 μmol · L$^{-1}$ Over the course of the 33-day experiment, the plankton community exhibited two distinct phases. In phase-I (days 5-20), a notable decline in the photosynthetic efficiency (F$_v$/F$_m$) was observed. This change was accompanied by substantial reductions in the abundances of picoeukaryotes, small size nanoeukaryotes (nanoeukaryotes-1), and microplankton. The cell viability of picoeukaryotes, as indicated by fluorescein-di-acetate hydrolysis by cellular esterases (FDA- green fluorescence), slightly increased by the end of phase-I whilst the viability of nanoeukaryotes 1 and *Synechococcus spp*. did not change. Reactive oxygen species levels (ROS-green fluorescence) showed no significant changes for any of the functional groups. In contrast, in phase-II (days 21-33), a pronounced community response was observed. Increases in F$_v$/F$_m$ in the intermediate OAE treatments of Δ900 to Δ1800 μmol · L$^{-1}$ and in chlorophyll-a (Chl-a), chlorophyll-c2 (Chl-c2), fucoxanthin and divinyl-Chl-a were attributed to the emergence of blooms of large size nanoeukaryotes (nanoeukaryotes-2) from the genera *Chrysochromulina*, as well as picoeukaryotes. *Synechococcus spp*. also flourished towards the end of this phase. In parallel, we observed a total 20% significant change in the metaproteome of the phytoplankton community. This is considered a significant alteration in protein expression, having substantial impacts on cellular functions and the physiology of the organisms. Medium levels of ΔTA showed more upregulated and less downregulated proteins than higher ΔTA treatments. Under these conditions, cell viability significantly increased in pico and nanoeukaryotes-1 in intermediate alkalinity levels, while in *Synechococcus spp*., nanoeukaryotes-2 and microplankton remained stable. ROS levels did not significantly change in any functional group. The pigment ratios DD+DT: FUCO, and DD+DT: Chl-a increased in medium ΔTA treatments, supporting the idea of nutrient deficiency alleviation and the absence of physiological stress. Taken all data together, this study shows that there is minimal evidence indicating a harmful impact of high alkalinity on the plankton community. The OAE treatments did not result in physiological fitness impairment, thus OAE did not cause cellular stress in the phytoplankton community studied.

Key words: Negative emissions technologies (NETs), carbon dioxide removal techniques (CDRs), ocean alkalinity enhancement (OAE), mesocosms, phytoplankton, chlorophyll a fluorescence, metaproteome, cell viability, reactive oxygen species (ROS), pigments, cell stress.

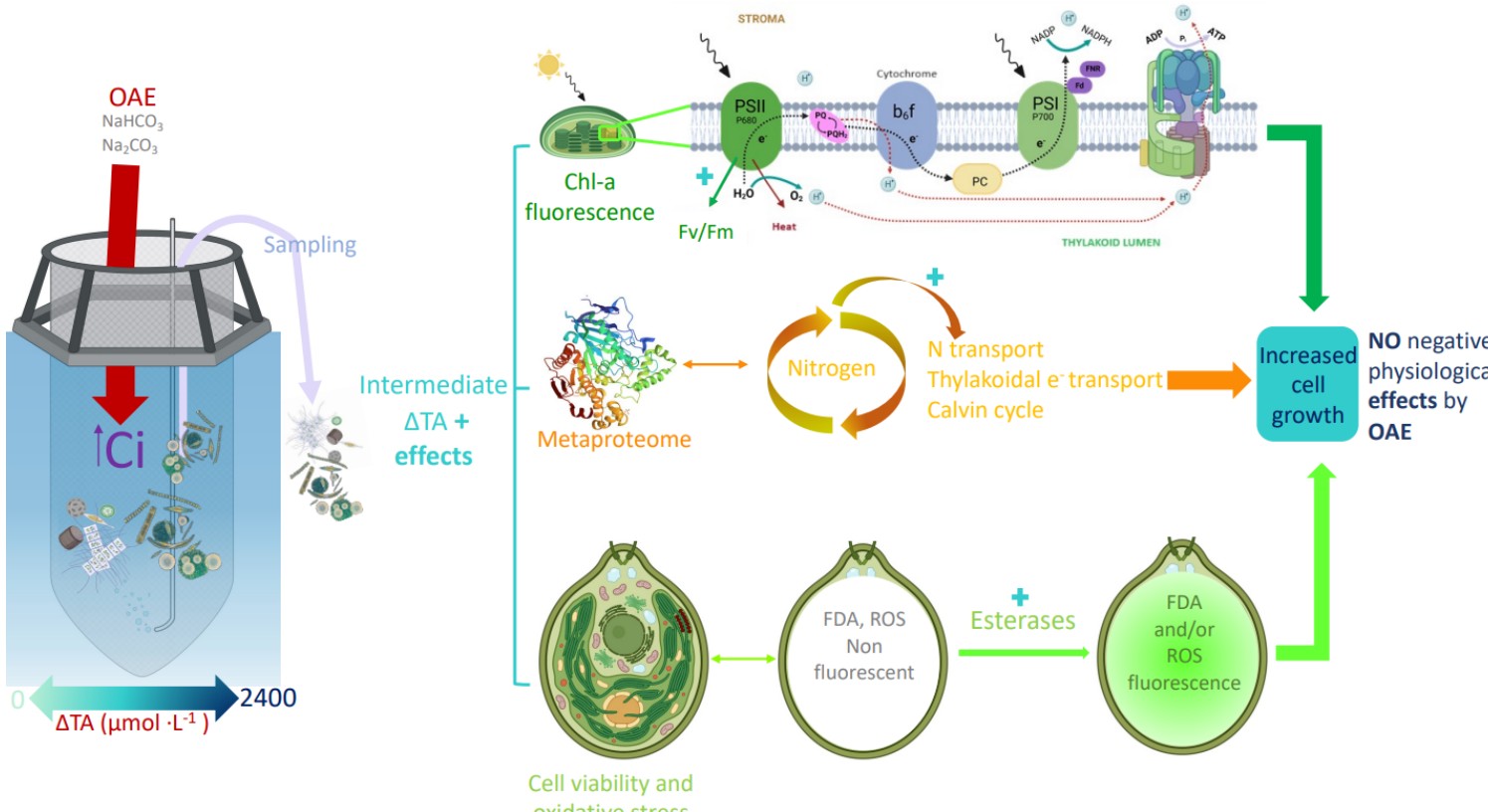

Ci: Inorganic Carbon

OAE: Ocean Alkalinity Enhancement

ΔTA: Alkalinity increments

**1 Introduction**

We are confronting one of the greatest challenges in the 21st century due to the progressive increase in atmospheric $CO_2$ caused by anthropogenic activities. The 2015 Paris Agreement represented a pivotal moment in global climate policy by setting the objective of keeping global temperature rise well under 2°C. This target was motivated by the urgency to prevent irreversible impacts and mitigate intolerable risks to human society. The Shared Socioeconomic Pathway (SSP) climate simulation-based scenarios (7AR IPCC 2021), combining elements from the new models about future societal development (SSP) with the previous iteration of scenarios and the representative concentration pathways (RPCs), describe new trajectories of changes in atmospheric greenhouse gases (GHG) over time, including very ambitious mitigation policies (Riahi et al, 2017). According to this, the most likely scenarios that we will be facing up to the year 2100 are the "middle of the road" (SSP2-4.5) and the "rocky road" (SSP3-7). The former is an intermediate GHG emission scenario, in which the predicted total cumulative $CO_2$ emissions will reach 600 µatm by 2100. In the latter, high GHG emissions may take place and $CO_2$ concentrations of 900 µatm are to be expected by 2100. This scenario also comprises a concomitant reduction in 0.25 units in the global ocean surface pH by 2100 with respect to 1950 levels for SSP2-4.5 and 0.35 units for SSP3-7, clearly leading to severe ocean acidification (OA). OA is already altering the carbonate system (consisting of the proportions of the different inorganic carbon forms maintained by the carbonate pump, the carbonate counter-pump, and the solubility pump in the ocean), that controls seawater pH (Millero et al. 2009). Carbon speciation is predicted to be highly affected, with an expected 17% increase in bicarbonate ions ($HCO_3^-$) and a 54% decrease in carbonate ($CO_3^{2-}$) concentrations in equilibrium. Thereupon, the saturation state of calcite ($\Omega_{calcite}$) and the rain ratio (RR, the ratio of calcite precipitation to organic matter production, i.e. PIC:POC ratio) will also change (Rost and Riebesell 2004, Doney 2020). The alteration of the carbonate chemistry in seawater due to a higher $CO_2$ will benefit some species such as nitrogen fixing cyanobacteria (Levitan et al. 2007) while being detrimental to others. OA negatively affects diatoms, prymnesiophytes and calcifier organisms (Riebesell and Tortell, 2011). Among the latter, OA highly impacts Haptophytes (e.g. coccolithophores). This phytoplankton group is of paramount significance in the global carbon cycle because it is responsible for marine biogenic calcification ($CaCO_3$ formation) and/or the rain ratio (Riebesell and Tortell, 2011). Therefore, OA influences community primary production, species composition, nutritive values such increase the C:N ratio and export, and the exchange of $CO_2$ between the surface ocean and the atmosphere (Riebesell et al. 2007).

Due to the impacts already caused and new predictions, it seems unlikely to achieve climate change mitigation aims without the help of ancilliary technologies that reduce atmospheric $CO_2$ concentration. In this sense, negative emissions technologies (NETs) (IPCC, 2021) include mechanisms such as CDRs (Carbon Dioxide Removal) that address the reduction and even elimination of atmospheric $CO_2$ (Riebesell et al., 2023). One potential CDR strategy involves increasing total alkalinity (TA) of the ocean, or ocean alkalinity enhancement (OAE). The consequent increase in $\Omega_{CaCO3}$ due to OAE would counteract ocean acidification by prompting proton uptake. This may also facilitate calcification (Feng et al., 2017), leading to an increase in the oceans carbon sinking capacity. I.e. OAE have the potential to increase the carbonate pump and the solubility

pump, both maintaining a surface-to-deep ocean gradient of dissolved inorganic carbon (DIC). Indeed, this type of NET consists of increasing the chemical storage capacity of the ocean for atmospheric $CO_2$, by adding proton-neutralizing substances to the surface ocean (Kheshgi, 1995). The subsequential alkalinity increase causes a change in the speciation of inorganic carbon in seawater, creating new space for the absorption of additional atmospheric $CO_2$ (Hartmann et al., 2013).

There are different methods of OAE to promote atmospheric carbon sequestration (Renforth and Henderson, 2017; Eisaman
et al., 2023; Bach et al. 2024). OAE can be achieved through the weathering of alkaline minerals like, limestone by the addition of carbonates (as hereby presented), silicate minerals such as the iron-containing mineral olivine, through electrochemical processes used to generate alkalinity from chloride brines or strong bases such as NaOH and by additions of hydrated-carbonate-minerals to seawater and ground minerals to the shelf seafloor.

However, we still lack an empirical foundation for a coherent comprehension of how the marine ecosystems will respond to
any of the OAE approaches that have been proposed since experimental data are scarce. Indeed, most of the available data are related to weathering experiments. Ferderer et al. (2022) found that using NaOH and $NaHCO_3$ as alkalinity treatments, OAE had significant but generally moderate effects on various groups in the phytoplankton community and on heterotrophic bacteria in natural coastal communities. In this experiment, more pronounced effects were observed for the diatom community where silicic acid drawdown and biogenic silica build-up were reduced at increased alkalinity. Hutchins et al.
(2023) exposed cultures of several phytoplankton groups to olivine leachates and found both positive and neutral responses but no evident toxic effects for the relevant ecological species *Nitzschia punctata, Ditylum bardawil, Emiliania huxleyi, Synechococcus spp, Trichodesmium erythraeum, Crocosphaera watsonii*. In both single and competitive co-cultures, silicifiers and calcifiers benefited from olivine dissolution products like iron, magnesium and silicate or enhanced alkalinity, respectively. The utilisation of olivine may contribute to a significant fertilization of seawater that could theoretically benefit
primary production.

Nevertheless, in the real-world experiments, olivine additions can have some drawbacks that need to be seriously considered. Experiments with this mineral have shown that the increase in alkalinity led to an increase in Ni 3-fold higher than the natural concentration in seawater (Montserrat et al., 2017), within the toxic range for many eukaryotic microalgae (Glass and Dupont, 2017). Other trace metals present in alkaline minerals, including, Fe, Cu, Cd, Cr, might impose further
ecotoxicological effects depending on their concentration in the minerals transferred to the water column or sediment (Beerling, 2017). By contrast, González-Santana et al. (2024) found in the very mesocosm experiment that we are here in discussing, that the introduction of carbon and sodium salts did not affect at all iron size fractionation, concentration, and iron-binding ligands, i.e. iron dynamics. Consequently, phytoplankton remained unaffected by alterations in this crucial element. Altogether, observed changes in phytoplankton communities suggest that a variety of physiological responses have
been predicted and observed in laboratory cultures or coastal phytoplankton communities. Yet, it remains challenging to

gauge what the response will be in more natural communities studied at sea, at larger scales and under more realistic scenarios.

The aim of the present work was to investigate the physiological response of different phytoplankton functional groups to OAE, using a mesocosm approach that simulated the environmental conditions following alkalinity additions and subsequent
equilibration of the carbonate system with the atmosphere. Based on the previous studies mentioned above, we hypothesised that we would not expect to see potential physiological consequences of increased equilibrated alkalinity, precisely due to equilibration. However, adding carbonate salts to the phytoplankton community could produce transient-quick acclimation processes in phytoplankton cells that most likely will display a reversible-stress response. Such a response might be mediated by nutrients (Falkowski & Raven, 2007). Hence, transcription of genes and the protein inventory will change accordingly. As
a result, the metabolic activity (i.e. cell viability), variable chlorophyll fluorescence (related to photosynthetic function), pigments composition and ratios (proxies for photoinhibition and physiological stress) and reactive oxygen species -ROS- accumulation (accounting for oxidative stress management) might also change depending on whether OAE exerts negative, positive, or neutral effects.
By studying this complementary suite of physiological response metrics, we hoped to gain a better understanding of how well-
suited natural phytoplankton cells are to thrive under OAE scenarios and whether this will subsequently translate into biological fitness. This constitutes a valuable tool for assessing the risks *versus* benefits of the effects of OAE on marine phytoplankton communities.

**2 Material and methods**
**2.1 Experimental design**

The mesocosm experiment was conducted in the port of Taliarte, Telde, Gran Canaria during a 33-day period from September to October 2021 (Fig. S1-supplemental material). These waters are characterised by warm surface temperatures and strong stratification of the water column, resulting in oligotrophic conditions, i.e., low nutrient content and low plankton biomass
(Arístegui et al., 2001). The sampling for the set of variables presented here was carried out every other day. A 7 L plexiglass cylinder, equipped with an integrated closure system to capture the water column without inducing stress to the organisms, was introduced into the mesocosm. Water samples from each mesocosm were quickly transported to the laboratory, maintaining the experiment's conditions.

The experimental design consisted of nine mesocosms (M1-M9) containing a natural water column of 8.3 $m^3$ each at 5 m depth, with a sediment trap attached to the bottom of the bag (Riebesell et al., 2013; Taucher, 2017). A simulation of air-balanced alkalinisation based on the addition of carbonates ($NaHCO_3$ and $Na_2CO_3$) was employed. On day 4 of the experiment, 40 L of natural seawater with different concentrations of carbonate salts previously homogenised, were added to each mesocosm, to achieve an alkalinity gradient between $\Delta 0$ (lowest) -$\Delta 2400$ (highest) $\mu mol \cdot L^{-1}$, reaching actual values of total

alkalinity (TA) on the seawater between 2300 and 4700 μmol · L$^{-1}$ · respectively. To illustrate this alkalinity gradient between mesocosms, in this manuscript and related studies in this special issue, a blue color gradient is used, with the lightest color corresponding to Δ0 and the darkest to Δ2400 μmol · L$^{-1}$, based on the pH colorimetric scale. Experimental design representing the alkalinity gradient (μmol · L$^{-1}$) in the nine mesocosms was M5 (Δ0), M1 (Δ300), M7 (Δ600), M4 (Δ900), M9 (Δ1200), M3 (Δ1500), M6 (Δ1800), M2 (Δ2100), M8 (Δ2400) μmol ·L$^{-1}$ (Fig. S1-supplemental material). For more details, see Paul et al. (2024).

## 2.2 Pigments

Chlorophyll a (Chl-a) and accessory pigments were analysed to discriminate between photosynthesis-related functions and cell stress or cell damage. The violaxanthin-anteraxanthin-zeaxanthin (VAZ) cycle and the diadinoxanthin-diatoxanthin (DD-DT) cycle may serve as indicators of activation of photoprotection mechanisms. The analysed pigments were Chl-c3, Chl-c2, total Chl-a, total divinyl-Chl-a, 19'-hexanoyloxyfucoxanthin, 19-butanoyloxyfucoxanthin, fucoxanthin, neoxanthin, violaxanthin, zeaxanthin, prasinoxanthan, peridinin, diadinoxanthin (DD), diatoxanthin (DT). Samples were analysed by reverse-phase high-performance liquid chromatography (HPLC; Van Heukelem et al., 2001) following collection by gentle vacuum filtration (< 200 mbar) onto glass fibre filters (GF/F Whatman, nominal pore size of 0.7 μm), with care taken to minimize exposure to light during filtration. Samples were retained in cryovials at −80 °C prior to analysis in the laboratory. For the HPLC analyses, samples were extracted in acetone (100 %) in plastic vials by homogenization of the filters using glass beads in a cell mill. After centrifugation (10 min, 10000 rpm, 4 °C) the supernatant was filtered through a 0.2 μm PTFE filter (LLG). From this, a Thermo Scientific HPLC Ultimate 3,000 with an Eclipse XDB-C8 4.6 × 150 mm 3.5 μm column was used to determine phytoplankton pigment concentrations.

## 2.3 Chlorophyll *a* fluorescence

An active chlorophyll fluorescence method was used to examine whether variations in the level of alkalinity enhancement, influenced photosynthetic physiology of the phytoplankton communities. Fast Repetition Rate fluorometry (FRRf) measurements were made using a FastOcean sensor (Chelsea Technologies Ltd, UK), either deployed directly in the mesocosms or attached to a FastAct module (Chelsea Technologies Ltd, UK) for discrete measurements at *in situ* temperatures. In each case, the FRRf single turnover sequences consisted of 100 x 1 s excitation flashlets on a 2 s pitch, followed by 40 x 1 s relaxation flashlets on a 50 s pitch. Each acquisition consisted of 60 sequences with an interval of 150 ms between sequences. For both *in situ* and discrete measurements, excitation flashlet intensity was set at 1.00 x 10$^{22}$ photons m$^{-2}$ s$^{-1}$ of blue light, centred at 450 nm. FastPro8 software (Chelsea Technologies Ltd, UK) was used to process the single turnover data using the biophysical model of Prasil et al. (2018). The background fluorescence, attributed to the instrument noise and fluorescent dissolved organic matter, was measured in 0.2 m-filtered seawater on each sampling date and subtracted from the fluorescence data using the FastPro8 software. To capture the dark-acclimated state of the phytoplankton communities, depth profiles of *in*

*situ* FFRf measurements were carried out pre-dawn. Acquisitions were set at 10 s intervals, so that ~ 10 acquisitions were obtained over the 2.5 m depth range in each mesocosm.

To investigate the photosynthetic capacity, light adaptation state and capacity to tolerate short-term changes in irradiance among phytoplankton communities at varied levels of alkalinity enhancement, discrete water samples, from the depth-integrated 5 l sampler, were used to conduct rapid light curves (RLC). Samples were kept in the dark for a minimum of 30 min prior to RLC measurements, to relax non-photochemical processes. The RLC protocol comprised 10 levels of actinic light ranging from 0 to 1052 mol photons $m^{-2}$ $s^{-1}$, with 30 s delay at the new light levels prior to light-acclimated, single turnover acquisitions and 10 s intervals between replicate acquisitions at each light level. A water jacket mounted on the FastOcean head kept samples at ambient water temperature during the measurements. The fluorescent terms derived from the single turnover acquisitions of dark adapted *in situ* or discrete samples are $F_o$, the fluorescence at t=0; $F_m$, the maximum fluorescence; and $F_v$ ($F_v = F_m - F_o$) the variable fluorescence. Their equivalents under light adapted conditions, $F'$, $F_m'$ and $F_q'$, respectively, were generated at each different level of photosynthetically active radiation (PAR) during the RLC measurements. Depth-averaged values of the parameter $F_v/F_m$, measured pre-dawn, provide an estimate of the photochemical efficiency of photosystem II (PSII) for the phytoplankton community in each mesocosm.

For the RLC analyses, relative PSII electron transport (rP, equivalent to rETR) was calculated as:

$$rP = PAR \times \frac{Fq'}{Fm'}$$

Photosystem II electron flux per unit volume was calculated using the absorption algorithm (Oxborough et al. 2012):

$$JV_{PSII} = \frac{Fq'}{Fm'} \cdot a_{LHII} \cdot E$$

Where $a_{LHII}$ is the absorption coefficient of PSII light harvesting and E is the incident photon irradiance. From the RLC of rP *versus* PAR and $JV_{PSII}$ *versus* PAR, the respective values of: $\alpha$, the initial slope; Ek, the value of PAR, at the inflection between $\alpha$ and the asymptote of the curve fit; and Pm, the light-saturated value of rP or $JV_{PSII}$ were derived from curve fits generated by FastPro8 software using two steps, an alpha phase to generate $\alpha$ and Ek, and a beta phase to estimate parameters beyond Ek, such as Pm (Webb et al. 1974, Silsbe and Kromkamp 2012, Boatman et al. 2019).

**2.3 Metaproteome analyses**

On day 27 of the experiment (t27), 2 L of seawater from each mesocosm was firstly firstly pre-filtered through 200 μm nylon mesh and then it was filtered through 0.2 μm polycarbonate filters (Whatman, Nuclepore, UK). Proteins were extracted from filters according to Segovia et al. (2003) and normalized to the same protein concentration. Thereafter, a gel-assisted proteolysis was carried out and digested protein samples were analysed according to the label-free quantitative LC/MS protocol according to Marrero et al. (2023). For this purpose, they were injected into an Easy nLC 1200 UHPLC system coupled to a

linear quadrupole-trap hybrid mass spectrometer-Orbitrap Q-Exactive HF-X (ThermoFisher Scientific). The raw acquired data were analysed using the Proteome Discoverer 2.5 platform (Thermo Fisher Scientific) with the Sequest HT and MASCOT search engines. The software versions used for data acquisition and operation were Tune 2.9 and Xcalibur 4.1.31.9. Peptides quantification was implemented using the Minora function of Proteome Discoverer 2.5. The following protein databases were used for the identification of proteins: *Synechoccocus* (UniProt, tax. ID: 1129); *Chrysochromulina* (Uniprot, tax. ID: 1460289); Microalgae (NCBI); Dynophyceae (NCBI); Bacillariophyta (UniProt); Protists (NCBI); Dyctyochphyceae (UniProt); *Emiliania huxleyi* (NCBI, tax. ID: 2903). The protein abundance values in each of the samples were normalized with respect to housekeeping proteins and scaled with respect to the control alkalinity treatment $\Delta 0$ $\mu mol \cdot L^{-1}$. The false discovery rate (FDR) for peptide and protein assignments was determined using the Percolator software package, imposing a strict 1 % FDR threshold.

## 2.4 Cellular viability and oxidative stress

Cell viability and oxidative stress were studied by using the cellular green fluorescence emission of specific probes (Invitrogen, USA) added to samples in vivo (Segovia & Berges, 2009). Organisms were size fractionated by filtering seawater through a 200 µm mesh followed by a 20 µm mesh. The fraction below 20 µm was used to analyse nanoeukaryotes (2-20 µm), picoeukaryotes ($< 2$ µm), and *Synechochoccus spp*. ($< 2$ µm). The fraction between 20- 200 µm corresponded to microplankton.

Cell viability was assessed with fluorescein diacetate (FDA). FDA is a nonpolar, non-fluorescent stain, which diffuses freely into cells. Inside the cell, the FDA molecule is cleaved (hydrolysed) by nonspecific esterases to yield the fluorescent product fluorescein and two acetates. Accumulations of fluorescein are the result of intracellular esterase activity, therefore indicating metabolic activity, i.e., cell viability. One mL sample was incubated with FDA at 20 µM final concentration, at 21 ºC for 2 h. Green fluorescence emitted by the cells was quantified with two flow cytometers Accuri C6 (Becton Dickinson, Belgium) and QuantaCell Lab (Beckman Coulter, Inc. USA). Two distinct flow cytometers were employed due to technical issues encountered with one of the instruments at the beginning of the experiment. Consequently, we have used the data from both instruments, after accurate intercalibrations. Counts were triggered using forward scatter (FSC) signals on the FITC channel. Reactive oxygen species (ROS) were assayed with carboxy-$H_2DFFDA$, a cell-permeable fluorescent indicator that is non fluorescent until oxidation by ROS occurs within the cell. $H_2DFFDA$ detects intracellular ROS species, and despite its lack of specificity, being oxidized by any possible radical with oxidative activity, it has been proven very useful and reliable for assessing the overall oxidative stress. One mL sample was incubated with ROS at 20 µM final concentration, at 21 ºC for 2 h. Green fluorescence was measured as described above. The percentages of stained cells and controls were calculated using FloJo_v10.8 software (Becton Dickinson, Belgium). Samples with FDA and ROS were analysed individually attending to the % of cells falling within the upper right FCM chart indicating green fluorescence *versus* the low right FCM chart indicating only red fluorescence.

The fraction between 20- 200 μm corresponding to microplankton was incubated with the fluorescent probes as described above and then filtered in darkness through black Anodisc/Nucleopore filters (25 mm Whatman) using a peristaltic pump at 0.2 mb to avoid cell damage and placed onto glass slides. A drop of immersion oil was infiltrated between the filter and the coverslip. Samples were then stored at 4 ºC until they were analysed by using an Eclipse E800 Nikon epifluorescence microscope under the FITC filter set. Red fluorescent chl-a cells were counted first, followed by the green fluorescent cells. The % of green fluorescent cells was calculated based on the total red fluorescent cell counts. A significant number of fields of vision (FOVs) were counted to have a significant N. The percentages of stained cells, positive and negative controls were calculated using Image J software.

It is important to note that fluorescent stains must be validated in each different experiment before use by analysing the optimal dye loading concentration and its kinetics for each specific phytoplankton community, to avoid sub-optimal fluorescence or saturated fluorescence reaching the flow cytometer's or epifluorescence microscope's laser detectors. This was done prior to the start of the experiment. In addition, negative controls for cell viability consisted of cells treated with 2 % glutaraldehyde (i.e. dead cells) (Segovia and Berges, 2009). Positive controls for cell viability consisted of cultured microalgae cells in log-phase and were provided by the Spanish Algae Bank (Taliarte, Gran Canarias, Spain). Positive controls for ROS (i.e. oxidised cellular components) consisted of cells treated with 0.05 % hydrogen peroxide, as indicated by Segovia and Berges (2009). General negative controls were done by substituting cells with both 0.2 μm filtered seawater and MilliQ water.

## 2.5 Microplankton analyses

Microplankton cell abundance was obtained using the FlowCAM8400, equipped with a colour camera, a 10X objective for 100X magnification, a FOV100 flow cell and a 1 mL syringe driven sample pump. A 125 mL sample from each mesocosm was collected every two days, and pre-filtered using a 280 μm mesh to exclude most of the zooplankton fraction. The samples were later preserved with a 1:1 solution of 37% formaldehyde and 100 % acetic acid (glacial). The Utermöhl method was employed by subsampling 50 mL for sedimentation into 6 mL, to concentrate cells (Utermöhl, 1958). The sedimented volume was aspirated, pre-filtered through a 200 μm mesh to avoid clogging and transferred to the FlowCAM. Following the recommended factory settings, the sample was run in auto-image mode with a flow rate of 0.1 mL · min$^{-1}$. Plankton and detritus particles were imaged at 14 frames per second. Planktonic functional groups (diatoms, dinoflagellates, silicoflagellates, protozoa) found within the target size range of 20-200 μm were sorted semi-automatically using the VisualSpreadsheet® Particle Analysis Software (Owen et al. 2022).

## 2.6 Statistical analyses

To examine the possible effects of the OAE treatments on the study variables, the data were tested for normality with the Shapiro-Wilks test, sphericity with the Mauchly test and homoscedasticity with the Levene test. All data met the requirements for parametric tests. Simple linear regression analysis ($p < 0.05$) was then used for all variables. To compare phase-I and phase-II of the experiment, t-tests ($p < 0.05$) were performed on FDA, ROS, and pigments. For cell viability and oxidative stress, split-plot ANOVAs were performed, followed by Bonferroni post-hoc test ($p < 0.05$). Statistical analyses were performed using the free software programs Jamovi (2022) and R (2022)

## 3 Results

To interpret the temporal responses, three distinct phases were defined according to variations in the seawater chemistry and the plankton community development. Phase-0 was established at the beginning of the experiment (t1-t3), representing the conditions in the mesocosms before the treatment was applied. Following this, Phase-I (t5-t20) commenced after alkalinization with NaHCO$_3$ and Na$_2$CO$_3$. Finally, Phase-II occurred from t21 to t33. The trends in alkalinity, dissolved inorganic carbon (DIC) and pH are illustrated in Fig. S2 in the supplemental material Indirect abiotic precipitation occurred in the highest treatment ($\Delta$2400 µmol · L$^{-1}$) lasting until the end of the experiment and leading to slight TA and DIC losses (Fig. S2-supplemental material). All nutrient levels measured were consistently low during the entire experiment (Fig. S3 - supplementary material) and there were no significant differences due to treatments. Silicic acid dropped off 1.5-fold between days 1 and 6 and remained stable around 0.2 µM (average) during the rest of the experimental period (Fig. S3A - supplementary material). Nitrate, nitrite and phosphate were under the detection limits of the analysis techniques employed (Fig. S3B, C - supplementary material). All this data can be consulted for further details in the article by Paul et al. (2024).

Total Chl-a initial values were ~0.5 µg · L$^{-1}$ up to t5, then it diminished to minimum values by the end of phase-I. Chl-a significantly increased in phase-II only in intermediate $\Delta$TA treatments (i.e. $\Delta$900, $\Delta$1500 and $\Delta$1800) between t21 and t33. Highest concentrations were achieved by t27 and ranged from 4 to 18-fold higher than in phase-I, with a final decline in all treatments (Fig. 1 and Fig. S5-supplemental material). The behaviour of the different plankton groups is summarized in section 3.1. and in Fig. S6-supplemental material. All this data can be consulted for further details in the article by Marín-Samper et al. (2024).

## 3.1 Community composition

During phase-I, picoeukaryotes significantly increased in $\Delta$600, $\Delta$900, $\Delta$1800 and $\Delta$2100 µmol · L$^{-1}$ treatments, as did the microplankton in the $\Delta$900 µmol · L$^{-1}$. On the contrary, the group nanoeukaryotes-1 followed a decreasing trend from the beginning of the experiment. Their maximal abundances were observed before the addition of the OAE treatments (t1-t3). In phase-II, nanoeukaryotes-2 dominated the community, specifically in $\Delta$1500 and $\Delta$1800 µmol · L$^{-1}$. Microplankton also increased by the end of this phase. The diatoms showed a decrease in cell numbers during all the experiment, except for M9 where they peaked twice in t18 and t23. Dinoflagellates presented a similar trend at the beginning on phase I. In phase II (t29-

31) all the mesocosms showed significant increase in cells numbers in M6 and M3 ($\Delta$1500 and $\Delta$1800, respectively). Silicoflagellates were only detected during phase II. *Synechococcus spp.* was the only functional group that increased its cell density over time in most of the mesocosms, mainly in the intermediate-low treatments reaching its maximum at t29 at $\Delta$600 µmol·L$^{-1}$ (Fig. S6-supplemental material) .

Pigmentary composition is used as a proxy to detect or confirm the presence of different functional groups in different assemblages over time. For instance, chlorophylls, fucoxanthins or divinyl-Chl-a serve as proxies to detect Haptophytes, Bacillariophyceae, dinoflagellates or cyanobacteria (Takaichi et al., 2011, Rodriguez et al., 2016, Ito et al., 2011). The pigmentary composition analysed by HPLC confirmed the flow cytometry data. We observed significant differences in pigments between phase-I and phase-II in some of the mesocosms (Fig. 1). By t27, intermediate $\Delta$TA treatments showed the highest total Chl-a, Chlorophyll-c2 (Chl-c2), fucoxanthin and divinyl-Chl-a, especially in $\Delta$TA 1800 Fig. 1G). DD, prasinoxanthin and neoxanthin showed a moderate accumulation in this mesocosm. Low and high $\Delta$TA treatments presented significantly lower concentrations than medium $\Delta$TA (Figs 1A-C, F, H). For instance, total Chl-a declined ~ 25-fold at the two ends of the alkalinity scale, compared to $\Delta$1800 at t27, as did fucoxanthin and divinyl-Chl-a concentrations. The only pigment that significantly increased during phase-I regardless of alkalinity, was peridinin (Figs 1B, D, H), which also slightly increased again later in phase-II at t25.

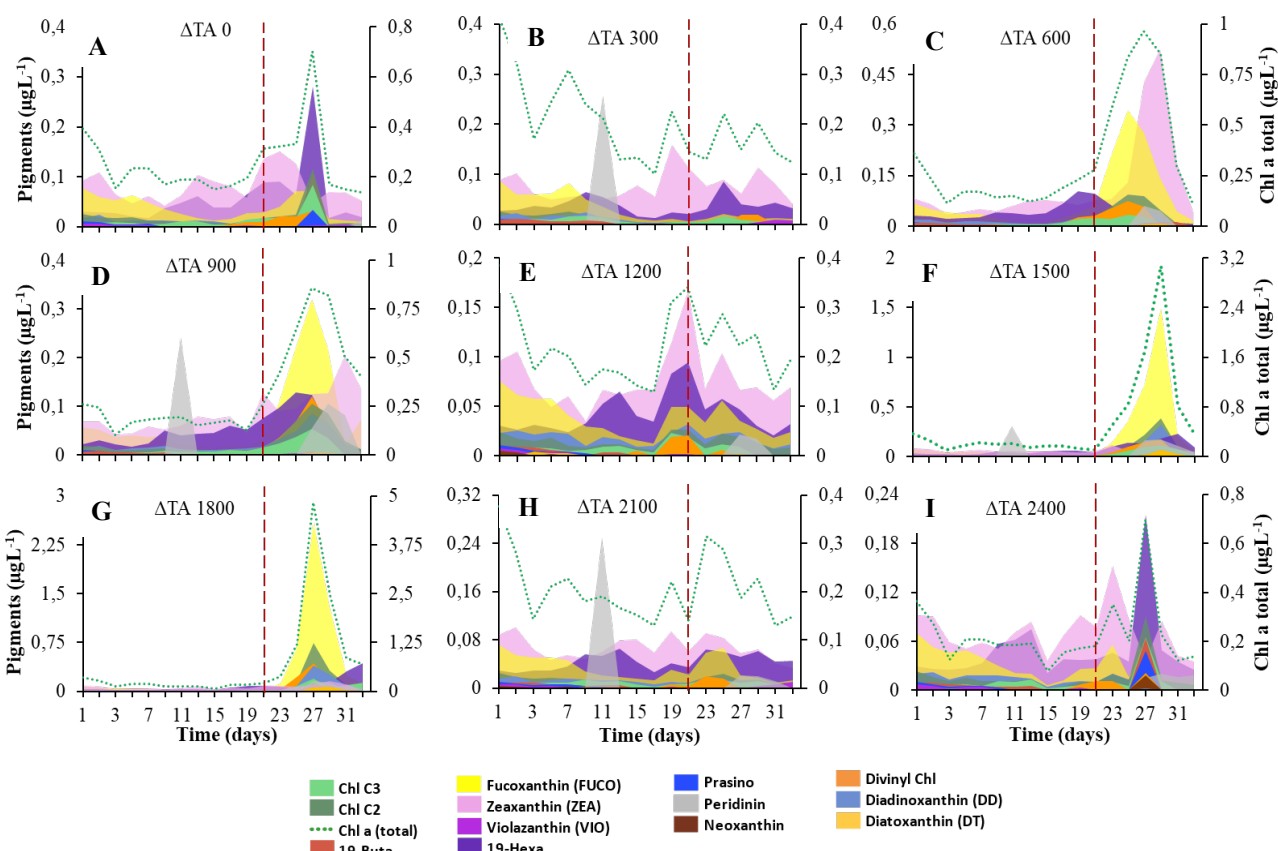

**Figure 1.** Cumulative concentration of accessory pigments in µg·L$^{-1}$ (primary Y-axis) and Chl a Total in µg·L$^{-1}$ (secondary Y-axis) by treatments. A) OAE 0 (M5), B) OAE 300 (M1), C) OAE 600 (M7), D) OAE 900 (M4), E) OAE 1200 (M9), F) OAE 1500 (M3), G) OAE 1800 (M6), H) OAE 2100 (M2), I) OAE 2800 (M8). Color codes for each pigment are included. The discontinuous red line separates phase I and phase II.

## 3.2 Phytoplankton physiological response

### 3.2.1 Photophysiological changes

Dark adapted PSII photochemical efficiency ($F_v/F_m$) of the phytoplankton communities initially exhibited a decline in all mesocosms, and for the remainder of phase-I between t7 and t20, stayed relatively constant, within the range of 0.25 to 0.38 (Fig. 2A). $F_v/F_m$ values were significantly different between levels of alkalinity and between phases, showing a highly significant interactive effect (Table 1). By the beginning of phase-II, $F_v/F_m$ was close to 0.30 in all mesocosms. At the start of phase-II, an increasing trend in $F_v/F_m$ and $\alpha$-JV$_{PSII}$ (Fig. 2C) was observed in mesocosms with intermediate alkalinity (up to the $\Delta$1800 treatment), with the highest $\Delta$TA mesocosms showing ~2-fold lower values. No significant differences were observed between only alkalinity levels in terms of the photosynthetic parameters $\alpha$-rP and $\alpha$-JV$_{PSII}$ during phase-I or phase-II (Fig. 2B, Table 1).

The increasing trend of the initial slope of electron flux per unit volume, α-$JV_{PSII}$, with increasing alkalinity that paralleled $F_v/F_m$ as it increased with OAE up to ΔTA1800, was not apparent in the statistical analysis for variations in only alkalinity but was apparent as a significant interactive effect of the phases and alkalinity (Table 1), possibly due to the discrepancy in response of the community at ΔTA 1200. In contrast to α-$JV_{PSII}$, the initial slope from the RLCs of relative PSII electron transport versus PAR, α-ETR, did show differences between phases but not between levels of alkalinity. The maximum rate of $JVPSII$ (PM) and non-photochemical quenching (NSV) did not present significant differences. Yet, NPQ shows an increasing pattern phase II in the intermediate treatments.

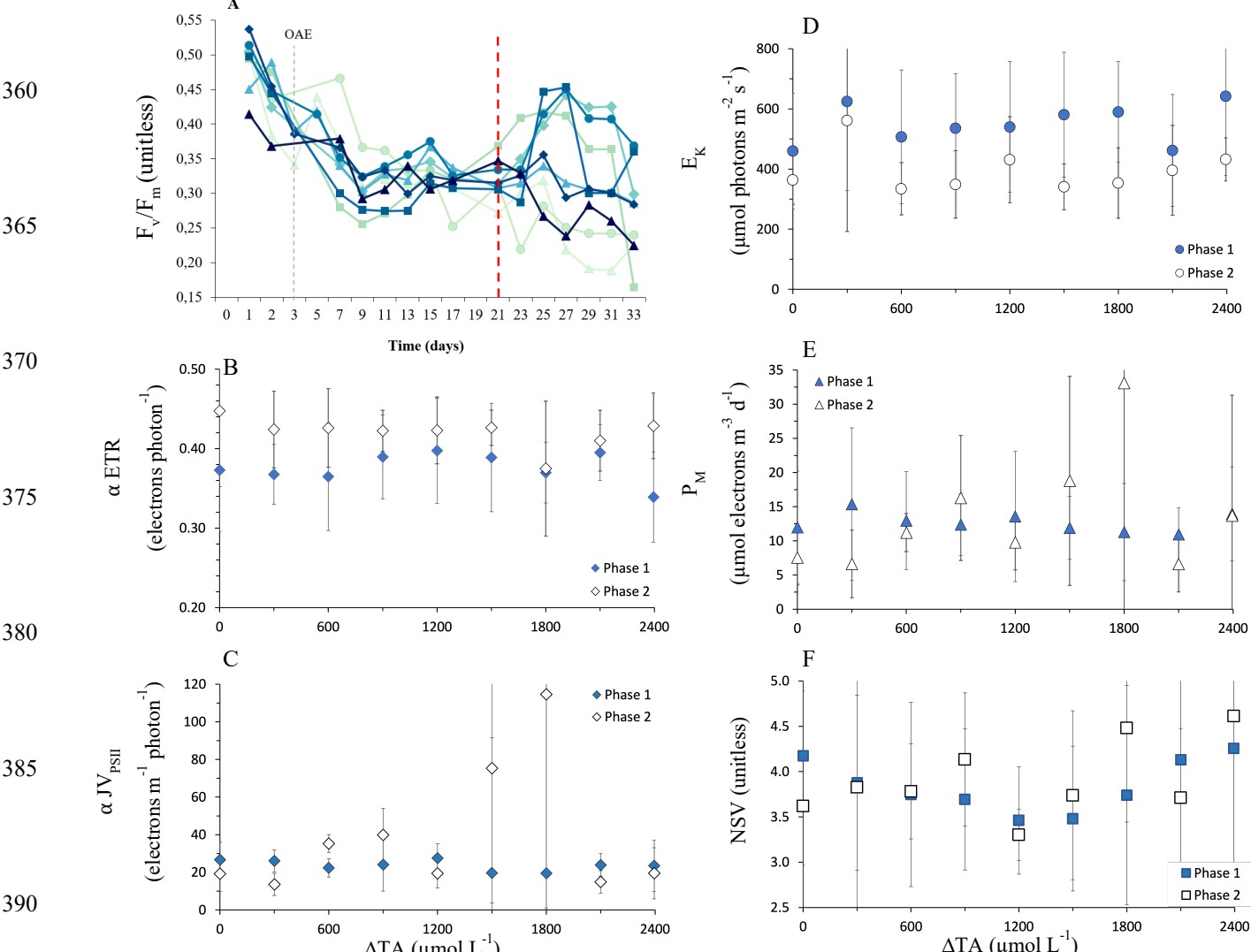

**Figure 2. A) Temporal trends in dark-adapted PSII photochemical efficiency (Fv/Fm). Values are averages of the pre-dawn measurements through the water column from each mesocosm. Discontinuos red line to differentiate between phases. Color gradient and symbols according to mesocosm alkalinity correspond to mesocosms with high alkalinity. B-D) Phase-averaged values of the Rapid Light Curve derived values with OAE of: B) α-rETR (relative**

electron transport rate); C) α-JVPSII (volume-specific electron transport rate); D) E$_K$ (light intensity for saturated JVPSII); E) P$_M$ (maximum rate of JVPSII); and F) NSV (non-photochemical energy dissipation or normalized Stern-Volmer quenching). Error bars are the SD of the Phase-averaged values.

400

Table 1. Statistical analysis of the effects of OAE between phase-I and phase-II of the experiment , as well as the differences among OAE treatments within each phase. Split-plot ANOVAs (i.e. mixed model ANOVA , general linear model (GLM)) followed by Bonferroni post-hoc test (p < 0.05), were performed on Fv/Fm, α-rP, α-JVPSII, FDA, ROS, and pigments. To analyse differences between treatments within phase-II for nanoeukaryotes-2 and microphytoplankton, split-plot ANOVAs with time as repeated measures factor were used followed by Bonferroni post-hoc test (p < 0.05). Statistically significant differences (p < 0.05) are indicated by an asterisk (*); ns: not significant (p > 0.05).

| | Differences between phases | Differences among OAE over time | Interaction OAE x Phases |
|---|---|---|---|
| **Variable** | *p-value* | *p-value* | *p-value* |
| **Photophysiology** | | | |
| F$_v$/F$_m$ | 0.034* | 0.010* | <0.001* |
| α-ETR | 0.002* | ns | ns |
| α-JV$_{PSII}$ | ns | ns | 0.009* |
| **Cell viability (FDA)** | | | |
| Picoeukaryotes | < 0.001* | < 0.009 | ns |
| *Synechococcus spp.* | ns | ns | 0.018* |
| Nanoeukaryotes-1 | < 0.001* | ns | ns |
| Nanoeukaryotes-2 | - | ns | 0.003* |
| Microphytoplankton | - | <0.001* | <0.001* |
| **Oxidative stress (ROS)** | | | |
| Picoeukaryotes | 0.004* | ns | ns |
| *Synechococcus spp.* | 0.001* | 0.009* | ns |
| Nanoeukaryotes-1 | <0.001* | ns | ns |
| Nanoeukaryotes-2 | - | ns | ns |
| Microphytoplankton | - | ns | 0.021* |
| **Pigments (HPLC)** | | | |
| 19-Hexanoyloxyfucoxanthin | <0.001* | ns | 0.004* |
| Zeaxanthin | <0.001* | ns | 0.029* |
| Fucoxanthin | 0.003* | 0.008* | 0.008* |
| 19-butanoyloxyfucoxanthin | ns | ns | ns |
| Diadinoxanthin | 0.001* | 0.002* | 0.003* |
| Chlorophyll c2 | <0.001* | 0.009* | 0.009* |
| Chlorophyll c3 | 0.006* | <0.001* | <0.001* |
| Divinyl Chlorophyll a (total) | <0.001* | 0.003* | 0.006* |
| Peridinin (total) | ns | ns | ns |
| Neoxanthin | ns | ns | ns |
| Prasinoxanthan | ns | ns | ns |
| Violaxanthin | ns | ns | ns |
| Diatoxanthin | 0.013* | ns | 0.011* |
| Chlorophyll a (total) | <0.001* | 0.008* | 0.004* |

410

*3.2.2 Metaproteome*

Fig. 3A depicts the study of 1750 proteins that were extracted and analysed. Volcano plots were used to test the statistical significance (-log 10 p-value) of the magnitude of change (log 2-fold-change) for a large dataset subjected to various conditions relative to the control. The X-axis represents the fold change, which indicates the effect of treatments on the abundance of a specific protein. On the Y-axis, values ≤ 1.30 for a p-value= 0.05 are considered nonsignificant. Thus, the higher the value is in the Y-axis the more significant is the change in abundance of a specific protein. Proteins that are downregulated appear on the left side of the plot, while upregulated proteins are on the right. We can then discriminate among the main tendencies of proteins that are subjected to significant changes in abundance depending on treatments.

Focusing on proteins that showed significant changes in abundance (Y-axis > 1.30), we observed differences between the mesocosms at high points of the alkalinity spectrum *versus* those in the middle. I.e. less downregulation and more upregulation occurred in mid-TA mesocosms compared to the higher TA treatments (Fig. 3, B and C). Δ1800 and Δ2100 μmol · L$^{-1}$ experienced 10 % and 12 % significant protein downregulation (Fig. 3B). Among the proteins with decreased abundances in these treatments were ATP synthase, rubisco (with high homology to diatoms and dinoflagellates), metal-binding proteins (similar to dinoflagellates), ribosomal proteins from prokaryotes, cytochromes (with high homology to dinoflagellates and diatoms), and exopeptidases. No proteins related to photosynthesis, nutrient acquisitions or cell growth seemed to be downregulated in these treatments.

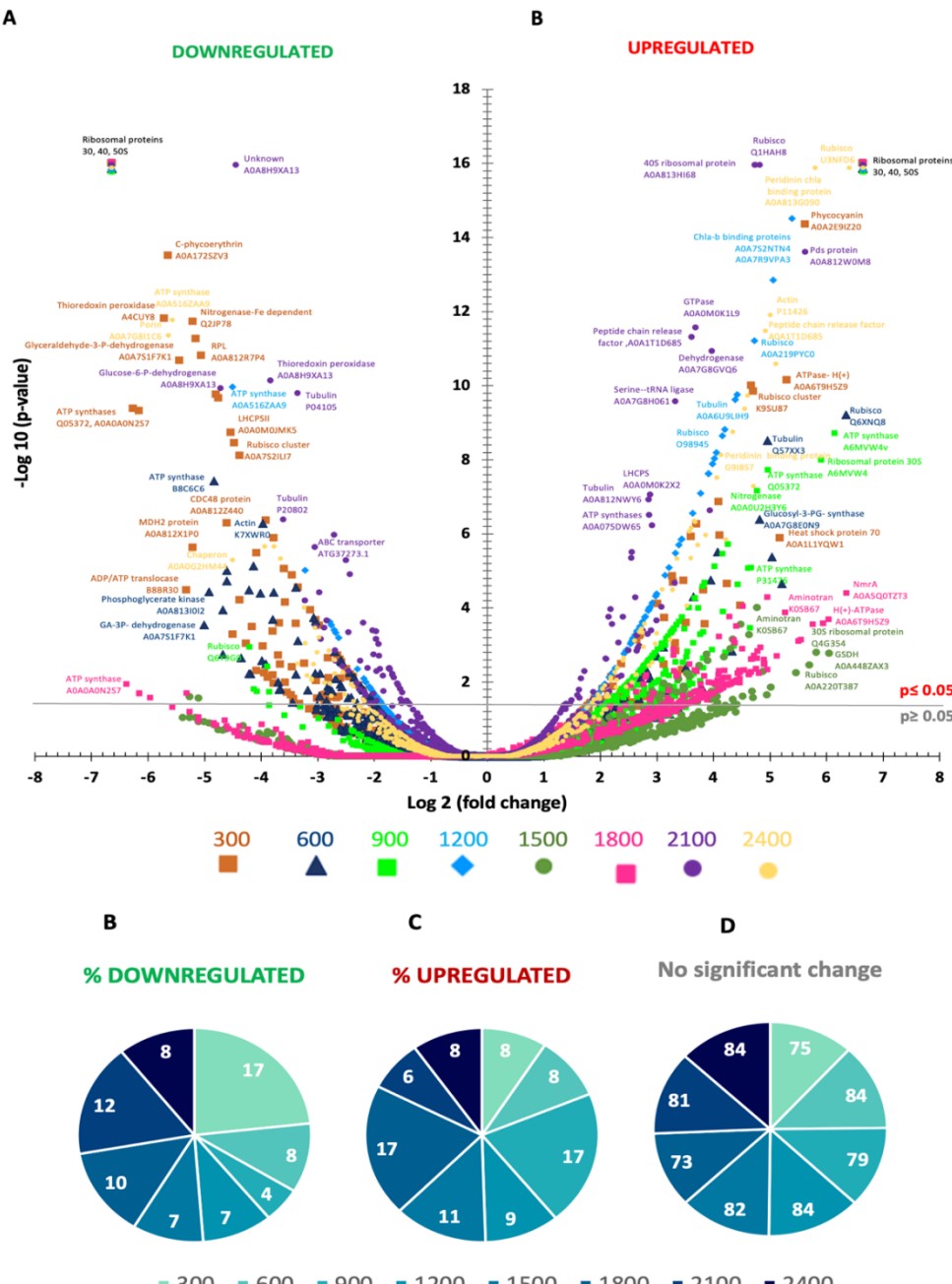

Figure 3. A) Volcano plot showing the metaproteome analysis on t27 of the experiment for all the mesocosms except for the control DTA 0. The statistical significance (-log 10 p-value) of the magnitude of change (log 2-fold-change) is tested with respect to the control. Each dot in the plot is one protein normalised by housekeeping proteins and by the control. Downregulated protein abundances appear in the left part of the plot, whilst upregulated ones are shown at the right half of the plot. B) Pie-sector plots representing the percentages of significant proteins abundances upregulated, downregulated and with no changes. Colour and symbol-codes for alkalinity levels for each mesocosm are shown below each plot.

Protein abundances dropped by approximately half in high and low treatments. The only exception of high downregulation in low TA treatments appeared to be $\Delta$300 μmol · L$^{-1}$, (17 %, Fig. 3B), probably due to high concentrations of ribosomal proteins downregulation, both from prokaryotes and eukaryotes. ATP synthases were also highly represented, with similarities to *Synechococcus spp*, diatoms, and Pedinellales. Chlorophyll-a and peridinin binding proteins followed in abundance, with high homology to dinoflagellates. Cytochromes of various types were also downregulated, with similarities to *Synechococcus spp*, dinoflagellates, and diatoms.

In the case of upregulated proteins (Fig. 3C), $\Delta$900 and $\Delta$1800 μmol · L$^{-1}$ exhibited a 17 % increase in abundances, many corresponding to ribosomal proteins and ATP synthases. Urea-ABC transporters were significantly increased, with high homology to Haptophyta and Cyanophyta, as well as Photosystems I and II, and rubisco. Chaperones and other proteins from the thylakoidal transport chain and Calvin cycle also contributed to the pool. $\Delta$1500 μmol · L$^{-1}$ showed an 11 % increase. The major percentages of accumulated proteins in this treatment were ribosomal proteins mainly from eukaryotes, chaperones, and AAA-ATP synthases. $\Delta$1200 μmol · L$^{-1}$ showed a 9 % increase consistently corresponding to ATP synthases, ribosomal proteins from eukaryotes and prokaryotes, PSII-associated proteins, and chaperones.

It is important to mention that the upregulated and downregulated data represent approximately 20 % of the total proteins. The significance of this 20 % will be further discussed in the following sections. Most of the proteins, averaging 80 %, did not exhibit changes in their abundances in response to the different $\Delta$TA treatments (Fig. 3A, D).

## 3.2 Specific functional group responses

### 3.2.1 Cell viability and cell stress

We analysed cell viability and ROS accumulation to study the physiological status of the community. During phase-I, averaged *Synechoccocus spp.,* picoeukaryotes and nanoeukaryotes-1 FDA-stained cells represented ~ 25 % each, of the total cell numbers (Fig. 4A, C, E). The two latter groups increased cell viability green fluorescence 2.2-fold in phase-II, evidencing significant differences between phases for the three functional groups (Table 1) (Figs. 4D, F). *Synechoccocus spp.* and nanoeukaryotes -2 only were significant in the interaction, indicating that the differences between phases and/or alkalinities were close to be significant (Table 1). Microplankton exhibited significant differences between treatments in phase-II. Unfortunately, technical issues prevented us from detecting microphytoplankton and nanoeukaryotes-2 using FDA and ROS fluorescent probes in Phase-I, therefore we cannot assess the differences between phases,

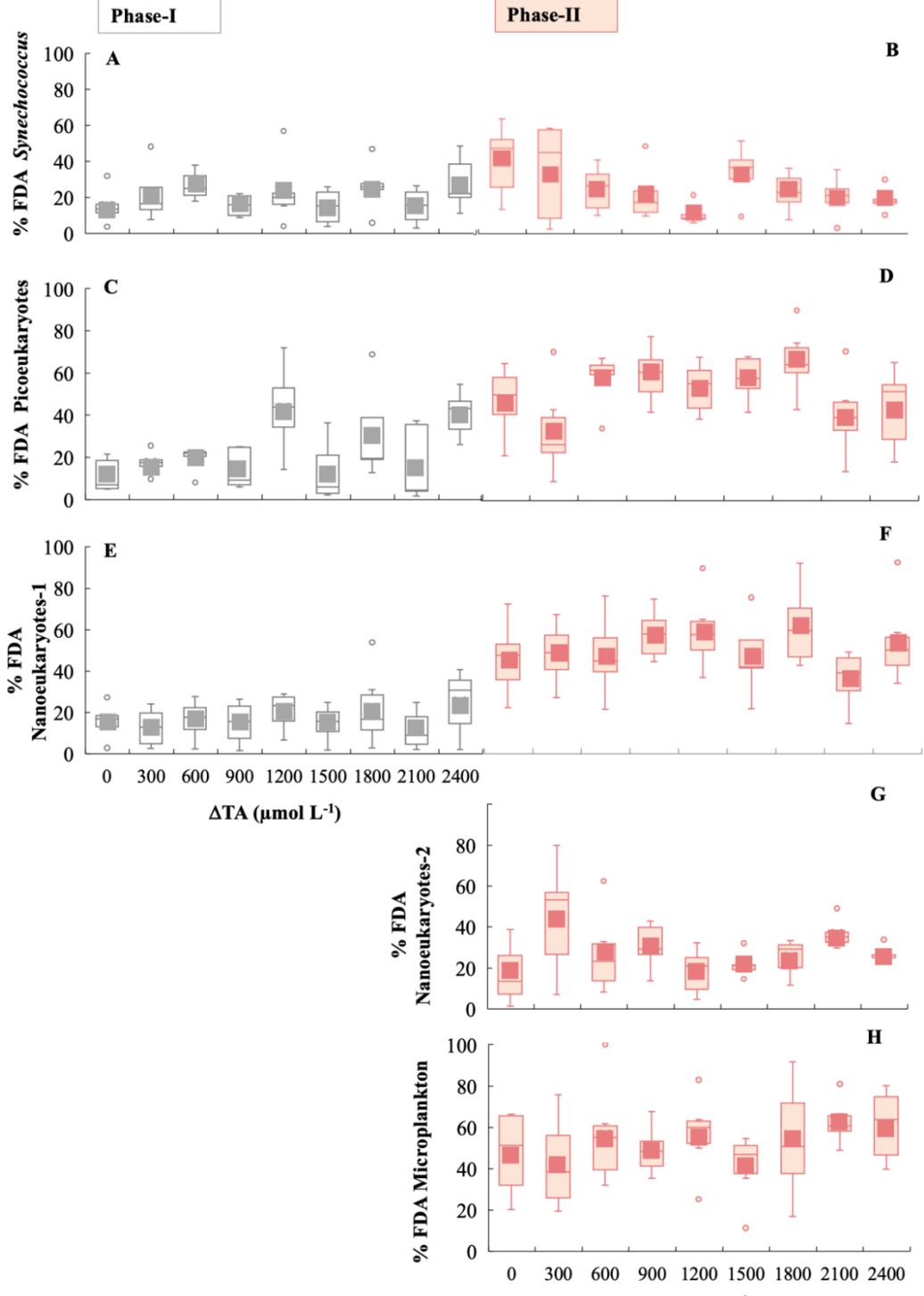

**Figure 4. Box and whisker-plots of the percentage of viable cells, determined using the FDA green fluorescence stain for phases-I and II respectively, for all functional groups. Grey color for phase I and salmon color for phase II. A, B) *Synechococcus* < 2 μm; C, D) picoeukaryotes < 2 μm; E, F) nanophytoplankton-1, <20μm; G) nanophytoplankton -2, > 20μm; H) microphytoplankton (30-280 μm).**

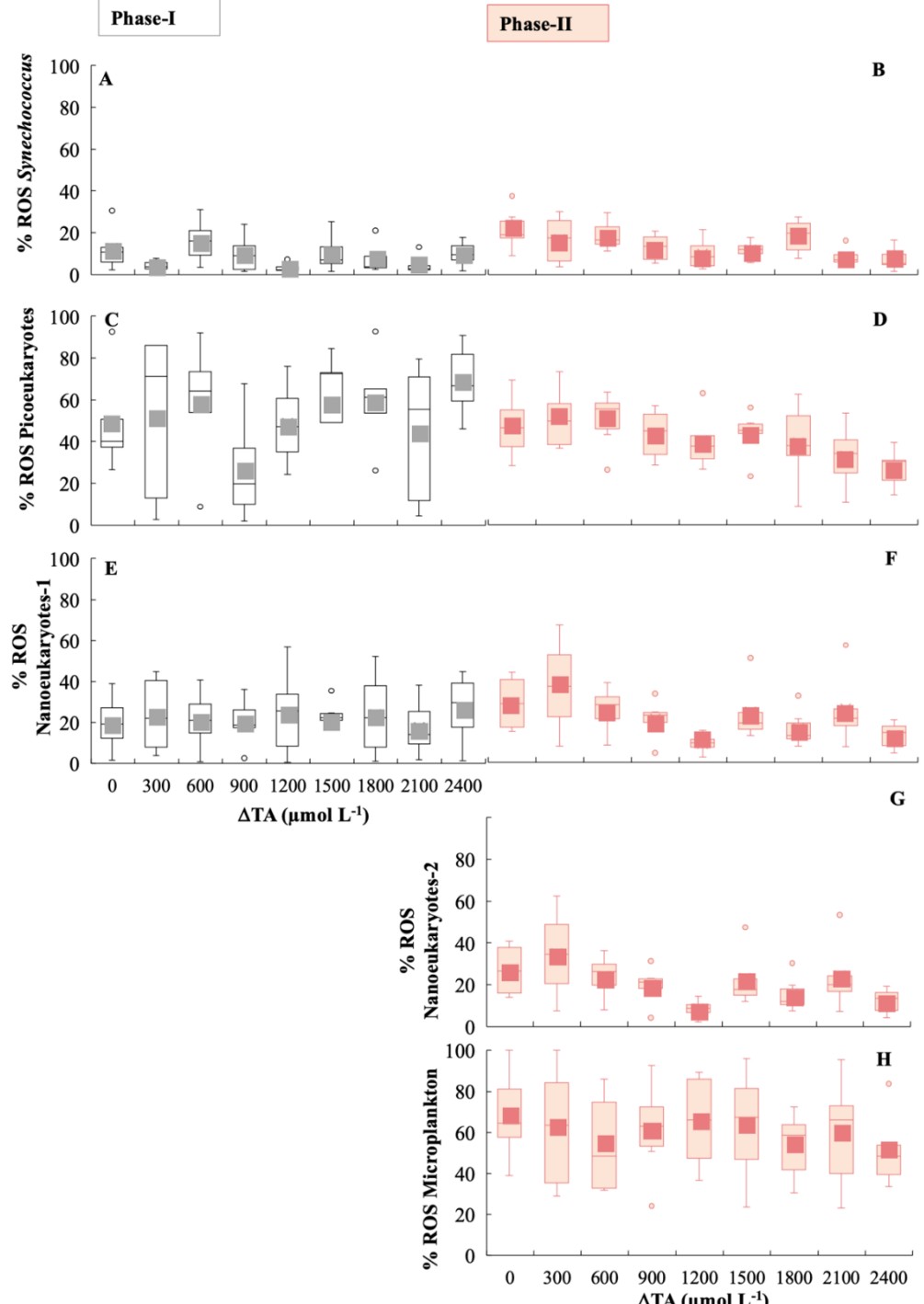

**Figure 5. Box and whisker-plots of the percentage of cells exhibiting the presence of ROS green fluorescence for phases-I and II respectively, for all functional groups. Grey color for phase I and salmon color for phase II. A, B) *Synechococcus* < 2 μm; C, D) picoeukaryotes < 2 μm; E, F) nanophytoplankton-1, <20μm; G) nanophytoplankton-2, > 20μm; H) microphytoplankton (30-280 μm).**

Averaged ROS fluorescence was 10 % for *Synechoccocus* (Fig. 5A), 45 % in picoeukaryote (Fig. 5B) and 23 % in nanophytoplankton < 20 µm in phase I. ROS accumulation in phase-II period seemed to slightly decline in medium TA levels for picoeukaryotes and nanoeukaryotes (Fig. 5D, F, G), the differences between phases being significant (Table 1). In contrast, none of the groups showed significant changes with respect to treatments except for *Synechoccocus ssp.* (Fig. 4B; Table 1).

### 3.2.2 Pigments related to the cellular physiological status

Nonphotochemical quenching (NPQ) is a regulated electron dissipation mechanism carried out via the xanthophylls cycles (VAZ and DD-DT), providing an indication of the phytoplankton communities' capacity for photoprotection (Raven, 2011). Pigments related to the xanthophylls cycle were analysed. They did not accumulate in significant amounts during phase-I. In phase-II violaxanthin (VIO) and zeaxanthin (ZEA) acting in the VAZ cycle, were detected although anteraxanthin was not. They did not accumulate much in medium ΔTA treatments except for ΔTA1200 (Fig 1D, F, G). However, these pigments especially increased in low and high ΔTA treatments during phase-II. VIO increased towards the end of phase-II (t31-33) in Δ300 and Δ2100 (Fig. 1I). ZEA peaked at Δ600 and Δ900 by t27. Diadinoxanthin (DD) and diatoxanthin (DT), pigments involved in the photoprotective de-epoxidation DD: DT cycle, followed the opposite pattern to VAZ pigments. They increased moderately in medium Δ1800 and Δ1500 treatments at t27 and t29 (Fig. 1F, G) but their concentration declined in low and high ΔTA mesocosms. In summary: 19-hexanoyloxyfucoxanthin, zeaxanthin, fucoxanthin, diadinoxanthin, chlorophylls c2 and c3, divinyl-Chl-a (total), diatoxanthin and Chl-a (total) showed significant differences between phase I and II. Fucoxanthin, chlorophylls c2 and c3, diadinoxanthin, divinyl-Chl-a (total) and Chl-a presented differences between OAE treatments and the interaction of most of them was also significant (Table 1).

The ratios DT+DD: TFUCO and DT+DD: Chl-a can be used as proxies of the cellular physiological status of the cells as well as nutrient limitation (Stolte et al 2003). DD+DT: TFUCO (mol: mol) remained constant with values below 0.3 mol: mol (Fig. 6A), except for Δ1500 at t27 and Δ900 at t31, when it significantly increased ($p < 0.05$). The DD+DT: Chl-a ratio experienced a continuous decline during phase-I (Fig. 6B) and a later recovery in intermediate ΔTA (i.e. Δ1800, Δ1500 and Δ900) except for Δ2100 during the last days of the experiment. Values for both ratios are presented from t19 onwards, as in phase-I of the experiment the pigment concentrations were close to zero.

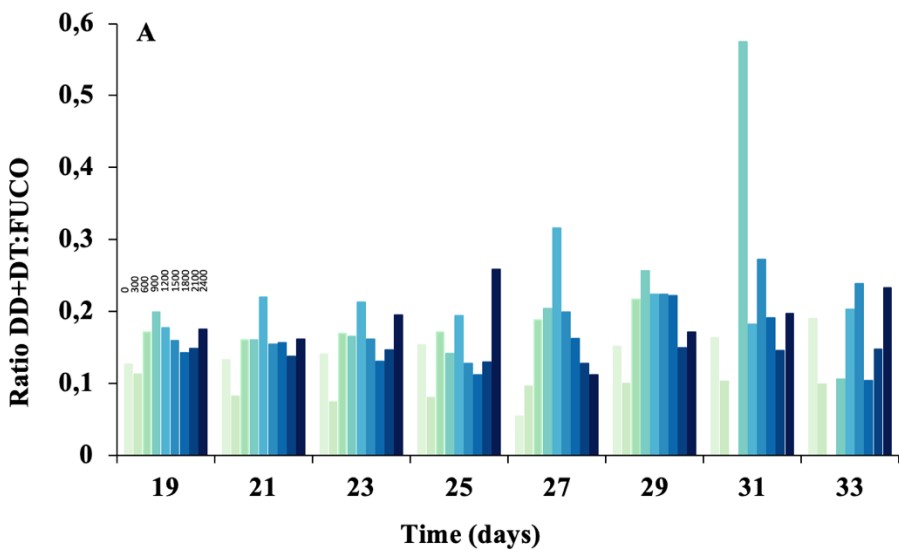

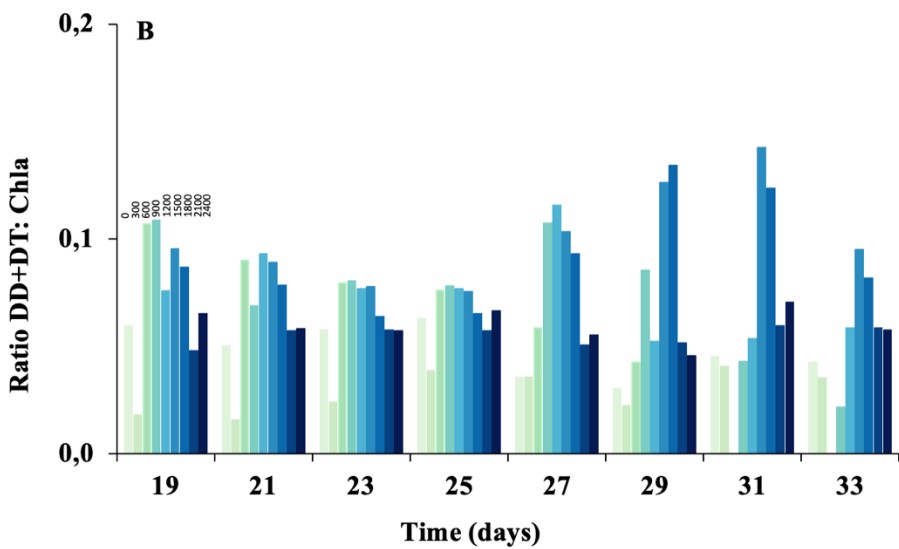

**Figure 6. Ratio of A) diadinoxanthin and diatoxanthin to total fucoxanthins (DD+DT: TFUCO) as a proxy for physiological stress; B) diadinoxanthin and diatoxanthin to chlorophyll _a_ (DD+DT: Chl-a) as a proxy for nutrient stress. Color gradient according to Fig.1.**

**4 Discussion**

The main goal of this study was to characterise the physiological status of the phytoplankton in each of the mesocosms to address whether there was a response within the communities to variation in the levels of equilibrated alkalinity enhancement.

One major concern regarding ocean alkalinity enhancement experiments is the stability of alkalinity over time since solid carbonate precipitation can occur when alkalinity is raised, due to the accompanying increase in $CO_3^{2-}$. As a result, surface ocean total alkalinity (TA) can decrease, provoking the opposite effect to the aimed target. To overcome such undesired outcome, $CO_2$ equilibrated solutions are applied to reduce the risk of precipitation. Our experimental setup effectively replicated an ocean liming scenario using a TA gradient increasing in 300 μmol · L$^{-1}$ intervals, while dissolved inorganic carbon (DIC) and TA remained constant, at least until day t21. On that day, in the highest treatment of Δ2400 μmol · L$^{-1}$, abiotic precipitation occurred (Marín-Samper et al. 2024), but this event was not expected to have influenced the phytoplankton community response as measured in this study.

## 4.1 Decrease in photophysiological fitness (phase-I)

The photophysiological fitness refers to the adequacy of cells to thrive in their environment by maintaining active photosynthesis, and it is different to the Darwinian concept of biological fitness which denotes an organism's ability to pass its genetic material to its offspring. Therefore, the direct assessment of the physiological fitness is given in our experiment for the phytoplankton community by variable fluorescence-based photochemical measurements and for specific functional groups by cell-specific fluorescence probes of cell viability and oxidative stress, as estimations of metabolic activity.

In Phase-I, the $F_v/F_m$ ratio started at 0.55 during t1. Subsequently, $F_v/F_m$ steadily declined across all treatments, coinciding with low chlorophyll-(Chl-a) concentrations. One of the causes for the decreasing photophysiological fitness during phase-I may be attributed to the dominance of heterotrophic processes within the community, a phenomenon commonly observed in the more unproductive regions of the ocean (Agustí et al. 2001), it may also be due to the drop in nutrient concentrations since the beginning of phase I, this depletion in inorganic nutrients may have benefited heterotrophic organisms. Nevertheless, considering that picoeukaryotes and small nanoeukaryotes were initially present at substantial concentrations, a more plausible explanation is that the values obtained for Chl-a and $F_v/F_m$ during phase-I reflected the community's overall low photosynthetic activity and low nutrients concentration. It is likely that confinement in the mesocosms lead to decreasing physiological fitness because of decreased nutrient availability at all levels of alkalinity.

A slight increase in the proportion of viable cells in picoeukaryotes and nanoeukaryotes-1 indicated by FDA green fluorescence at the end of phase-I, suggests populations with an increasing capacity for growth. However, these group's cell numbers were low in the mesocosms, which is consistent with a decline in $F_v/F_m$ for the bulk phytoplanktonic community. Cell viability serves as a direct measure of metabolic activity in phytoplankton since the emission of green fluorescence, following the cleavage of fluorescein-diacetate molecules by cytoplasmic esterases, is associated with the basal metabolism. Metabolic (esterase) activity remains in steady state or increase when cells are well acclimated and not affected by stressful or disturbing conditions. The opposite occurs when one or more drivers exert effects on the cells (Sobrino et al. 2014). The reason for

diminished FDA fluorescent cells could be nutrient limitation, metabolic downregulation, photoinhibition, or a combination of all (Harrison & Smith, 2013). Certainly, there was nutrient limitation due to the oligotrophic conditions of the Atlantic waters (Paul et al. 2024). Studying different phytoplankton species Garvey et al (2007) demonstrated that, even with well-tested species, transient phases of apparently low cell viability may occur during times of physiological changes, e.g. due to nutrient limitation, aggregation, and any environmental stress. Consequently, alive cells with low esterase activity can easily be mistaken with non-viable or dead cells although they may recover and resume metabolic activity again when the stress ceases.

The low FDA green fluorescence was possibly due to the downregulation of carbon concentration mechanisms (CCMs) by increased DIC levels (Giordano et al. 2005, Sobrino et al. 2014). The CCMs function to actively transport dissolved inorganic carbon (Ci) species within the cell that is then utilised to provide an elevated $CO_2$ concentration around Rubisco and are activated when Ci is limiting. This was not the case in our study because DIC was maintained steady at high levels in each mesocosm along the entire experiment (Fig. S2-supplemental material). For instance, it has been reported that TWCA1, a cluster of carbonic anhydrases found in numerous marine phytoplankton species, exhibits both $CO_2$ hydration and esterase function (Lee et al. 2013), signifying that some esterases function in phytoplankton is closely linked to the CCMs. However, if these circumstances would have applied, cell viability would have been unlikely to increase during phase-II due to the maintained high DIC levels. Our results seem to align with the idea of esterase function being contingent upon a basal metabolism. But still, basal metabolism due to downregulation or due to restricted cell growth? It is conceivable that during phase-I, low levels of FDA (~ 25 %) were detected even though the cells remained alive and viable, because the cells were undergoing a physiological deficit, which in turn hindered their growth and slowed-down their proper acclimation, and not due to inhibition of CCMs leading to metabolic downregulation.

Other possible scenario that may curtail metabolic activity is photoinhibition. This typically entails a reduction in the rate of photosynthesis, the thylakoidal electron transport chain, and changes in pigment composition because of excessive irradiance and/or high ROS accumulation (Falkowski and Raven, 2007). The decrease in ROS agrees with the premises discussed just above since they are a direct by-product of highly active photosynthesis, and [14]C-fixation dropped off in this period (Marin-Samper et al., 2024) due to a lowered metabolism. The concentration of accessory pigments, specifically the decrease in oxygenated carotenoids from the VAZ cycle and the unchanged concentration of DD and DT in phase-I coupled with the non-significant slight decrease in photosynthetic efficiency, suggest that cells were not photoinhibited in any of the mesocosms (García-Gomez et al. 2016; Segovia et al. 2018). In *Synechococcus spp.*, cell viability and ROS exhibited a steady-state pattern, independent of the prolonged upward trend in cell numbers during Phase-I. It is well-documented that cyanobacteria metabolism exhibits niche differences (Desai et al. 2012), and the mechanisms regulating the growth of *Synechococcus* are further discussed within the next section.

Taking all factors into consideration, we suggest that during phase-I, depleted $F_v/F_m$, cell viability and ROS levels reflected the low basal metabolism in the functional groups that were present. Under this scenario, active photosynthesis was sustained as much as possible in a nutrient-deficient environment, rather than being associated with any cellular stress induced by $\Delta$TA. The non-significant regressions analyses, support that $\Delta$TA did not exert effects on the cell stress variables according to our starting hypothesis because of the equilibrated state.

## 4.2 Changes in the community (phase-II)

Phase-II occurred between t20 and t33, and several noteworthy changes were observed. Chlorophyll-a (Chl-a) levels increased, in parallel with a significant recovery in the $F_v/F_m$ ratio, and raised $\alpha$-JV$_{PSII}$, and $\alpha$-rP values, particularly in treatments with
medium $\Delta$TA ($\Delta$900, $\Delta$1500 and $\Delta$1800 µmol · L$^{-1}$), apart from $\Delta$1200. Conversely, $F_v/F_m$ values dropped-off in the low and high $\Delta$TA mesocosms during this phase. As mentioned earlier, variations in $F_v/F_m$ can be indicative of stress directly affecting Photosystem II (PSII). Excess irradiance injures phytoplankton by inducing photobleaching of photosynthetic pigments and by altering complexes and molecules present in the photosynthetic apparatus such as the Photosystem II (PSII) complex and/or the rubisco enzyme (Häder et al. 1998) due to overshooting electrons and oxy-radicals (Segovia et al. 2015). However, the
cells react to high light conditions by triggering photoprotection mechanisms (Segovia et al. 2015). Pigments serve as a proxy because some of them are related to photoinhibition and/or photodamage processes. The xanthophyll cycle (oxygenated carotenoids) is directly related to non-photochemical quenching (NPQ). On the other hand, the violaxanthin-antheraxanthin-zeaxanthin (VAZ) cycle has been described in chlorophytes, and the diadinoxanthin-diatoxanthin (DD-DT) cycle is present in Bacillariophyceae, Haptophyta, and in most algae that contain chlorophyll (Takaichi 2011)."
Our data suggests that under moderate $\Delta$TA conditions, the community experienced a relaxation in its photosynthetic state since Fv/Fm increased, indicating the increasingly effective transfer of electrons to photochemistry, without causing the accumulation of reactive oxygen species (ROS). Such relaxation is further supported by the lack of an increase in accessory pigments (Fig. 1) and the behaviour of the DD+DT: TFUCO and DD+DT: Chl-a (mol: mol) ratios (Fig. 6). In multi-species bloom situations, the ratios of DT+ DD:TFUCO and DT+DD:chl-a can be used as an indicator of the cellular physiological
status (Stolte et al. 2000). Indeed, what in part determines phytoplankton species fitness, succession and distribution are the physiological trade-offs between light harvesting, photoacclimation or photoprotection and dissipation of excess energy, including oxy-radicals, in which all pigments participate. This is reflected by changes in the pigment ratios of DT+DD:TFUCO and HEXA:TFUCO indicating physiological stress (Stolte et al. 2000). Upon examining DD+DT: TFUCO, we observed a significant increase of this ratio in intermediate $\Delta$TA. Fucoxanthins serve as a highly efficient light-harvesting pigment when
irradiance is optimal (Harris et al. 2009). The lower these ratios are, the more stress is exerted upon the cells. For instance, a lowered (DT+DD): TFUCO (mol: mol) ratio implies a dilution of the DD−DT cycle, meaning that the cells' capacity for non-photochemical quenching (NPQ or NSV as denoted in Fig.2) is decreased, being prone to excessive ROS and photodamage (Segovia et. al 2018). NSV increased in phase II coincidentally with bloom peaks in the intermediate treatments (Fig. 2F). In

parallel, increases in DD+DT:TFUCO were observed . A higher NPQ means that the capacity of the cells to dissipate excess energy is enhanced, thus we can exclude photodamage as a direct responsible for $F_v/F_m$ declines. Additionally, no differences were observed between treatments suggesting that the OAE treatments did not negatively influence the photophysiological state of the phytoplankton community. It is important to note that all these pigments are involved in the long-term photoresponse (Dimier et al. 2009), explaining their absence during phase-I. Along this line, the DD+DT: Chl-a ratio also might as well have increased in intermediate ΔTA due to a possible input of nutrients

However, the above discussed pigment concentrations and pigment ratios-functionality might not be the only explanation as to why $F_v/F_m$ and in general all the photobiology parameters changed during the experiment with different ΔTA. Pigment content can vary based on taxonomic shifts as well as on physiological adjustments (Aiken et al. 2009). These significant improvements in Chl-a, Chlorophyll-c2 (Chl-c2), fucoxanthin and divinyl-Chl-a containing species as well as in photophysiology data during phase-II, are ascribed to the emergence of blooms of nanoeukaryotes-2 from t23 to t27 when they peaked. Specifically, Haptophytes from the genera *Chrysochromulina* in Δ1500 and in Δ1800. Picoeukaryotes also peaked in this period in such mesocosms. The increase of zexanthin (ZEA) in ΔTA 600 matched the significant growth of *Synechococcus spp* . ZEA also increased in ΔTA 1200 when higher diatoms numbers were present at the same time with the nutrients, because there was an increase of silicic acid (Fig. S3, A-Supplemental Material). Fucoxanthin rising seemed to be related with silicoflagellates increases in ΔTA 1800, and peridinin elevation was related to dinoflagellates but in phase-I.

There are several studies suggesting that the values of $F_v/F_m$ of natural phytoplankton are largely determined by the algal taxa present (particularly those contributing the most to total Chl a) and by changes in composition over time, i.e. taxonomic-dependence of $F_v/F_m$ (Suggett et al., 2004, 2009). This is indisputably part of the explication of the changes observed in our experiment. However, there is another component to it. Some authors use the $F_v/F_m$ parameter as" photosynthetic competence", assessing the behaviour on phytoplankton regarding nutrient availability (Li et al. 2015). I.e. $F_v/F_m$ is enhanced when nutrients are supplied to oligotrophic communities and total chl-a increases in turn, enhancing the "photosynthetic competence" of some functional groups over the other. We do concur with this sense of $F_v/F_m$ since as explained somewhere above in this manuscript, the photophysiological fitness refers to the adequacy of cells to thrive in their environment by maintaining active photosynthesis. In this case, within an environment that was oligotrophic in origin.

It is indeed somehow striking that given the high oligotrophy of the system, phytoplankton (*Chrysochromulina* in Δ1500 and in Δ1800) flourished during phase-II. Metabolism drives traits that determine fitness, growth, and survival of populations; thus, one may wonder what may have helped the cells to meet the extra metabolic demands imposed by limiting nutrients and increased TA, allowing them to sustain/increase growth, especially during this phase-II. Unfortunately, neither $NH_4^+$ nor heterotrophic bacterial production were measured in this experiment, and these factors may hold the key to understanding the phytoplankton growth dynamics. However, heterotrophic bacterial numbers did increase in this phase (unpublished data), as well as DON and DOC.

We could speculate that ammonium ($NH_4^+$) increased in phase-II, likely because of remineralisation and nutrient release from the decaying phytoplankton, specifically nanoeukaryotes -1 and microphytoplanktonic diatoms and dinoflagellates during phase-I. In spite that most of the phytoplankton are osmotrophs (Paulino et al. 2008), phytoplankton using $NH_4^+$ for growth have reduced metabolic demands than phytoplankton using organic compounds and nitrate ($NO_3^-$), because $NH_4^+$ can be directly incorporated into amino acids (Maldonado & Price 1996, Schoffmann et al. 2016). Henceforth, the ability of phytoplankton to thrive at low $NO_3^-$ levels can be explained by its ability to efficiently use $NH_4^+$ and organic nitrogen sources, as observed in the North Sea and in Norwegian fjords (Lessard et al. 2005, Segovia et al. 2017). McKew et al. (2015) reported that for example coccolithophores can use N sources other than nitrate highly efficiently, by increasing the abundance of $NH_4^+$ transporters under nitrate limitation.

We observed a significant total change of 20% in the proteome. This is considered a significant alteration in protein composition. Proteome changes, especially of that magnitude, can have a substantial impact on cellular functions and the physiology of an organism. Therefore, it shall be regarded as a substantial shift in protein expression having significant biological implications (Goemine et al. 2018). Our metaproteome study supports the concept of cells benefitting from $NH_4^+$ during phase-II, based on the finding of urea-ABC transporters being highly upregulated in most of the mesocosms by t27 (Fig 7A, Fig. S9-Supplementary material). Urea is the direct precursor of $NH_4^+$, and many phytoplankton groups exploit this molecule (Baek et al. 2008). ABC transporters are one of the largest and possibly one of the oldest transport systems superfamilies (Thomas, 2020). They are represented in all extant phyla, from prokaryotes to humans, and they consist of multiple subunits, which are transmembrane proteins, and membrane-associated AAA-ATPases. Coincidentally, AAA-ATPases were also spotted in the metaproteome to be highly upregulated in most of the mesocosms on t27 (Fig. S9-Supplementary material), further evidence reinforcing the assumption of phytoplankton utilising $NH_4^+$. The proteome analyses also revealed that on t27, 7% of protein abundances were downregulated, and 11% were upregulated in $\Delta$1500 µmol · L$^{-1}$. For $\Delta$1800 µmol · L$^{-1}$, 10% were downregulated, and 17% were upregulated. Notably, the biological functions and processes associated with these proteins were similar in both $\Delta$TA treatments. The upregulated protein abundances corresponded to functions such as chloroplast ATP synthesis (ATP synthases), eukaryote mRNA translation (ribosomal proteins), protein folding during translation (chaperones), carbon fixation (rubisco), mitochondrial oxidative phosphorylation (cytochromes), Photosystem II turnover (reaction centre proteins), and N-transport into the cell (urea-ABC transporters), all of which belonged to both the Eukarya and Prokarya kingdoms. Among the proteins with the highest downregulated abundances were ribosomal proteins of prokaryotic origin. Additionally, there was a decrease in light-harvesting processes (peridinin-like chlorophyll-a/b-binding protein), and in physiological stress management (heat shock 70). The proteome data supports the evidence of active growth and biomass increase in phase-II at intermediate levels of alkalinity enhancement, without the development of physiological stress.

Cell viability increased 2.2-fold in picoeukaryotes and in nanophytoplankton < 20µm duirng phase-II. However, while picoeukaryotes increased in cell numbers during this period, cell density decreased in the latter group. This inverse correlation between FDA fluorescence and the trend in cell numbers could possibly be due to high grazing pressure on highly viable cells.

Unfortunately, despite the occurrence of blooms and presumably higher growth rates for nanoeukaryotes > 20 µm, and microplankton during phase-II, we cannot assess whether their cell viability and/or ROS content changed with respect phase-I since we were not able to measure these variables because of technical issues. *Synechococcus spp*. dominated the abundance of the picoplankton community (Fig. S6-supplemental material) and their increasing cell abundance during phase-II also contributed to the restoration of Chl-a and $F_v/F_m$, particularly in medium $\Delta$TA treatments. Non-significant differences were

observed between phases for both FDA and ROS green-fluorescence. This species distribution displayed a slow but consistent growth rate from the onset of the experiment until it peaked at the end. We were able to distinctly identify two clusters of *Synechococcus spp*. based on their fluorescence characteristics in the flow cytometry analysis. This distinctness in clusters were probably the result of the variations in light acclimation.

Proteome analyses unveiled a substantial upregulation of the *Synechococcus spp*. rod-linker polypeptide and allophycocyanines (Fig. S9-Supplementary material) which are integral components of the phycobilisomes, a sophisticated light harvesting system particularly advantageous deeper in the water column described for cyanobacteria (Adir et al., 2020). On the other hand, it is possible that *Synechococcus spp*. CCMs were downregulated since cell density and Fv/Fm increased during phase-II, but ROS fluorescent cells decreased, and FDA-stained cells did not vary. The genes for several CCMs

transporters are genetically induced under Ci limitation via transcriptional and translational regulatory processes (Price, 2011). Nevertheless, we did not find any CCMs in the metaproteome study, confirming that precisely Ci was in excess. The ultimate aetiology for low cell viability can be on one side physiological deficit as in phase-I, and in the other, metabolic downregulation due to luxury Ci in phase-II, explaining the absence of increased FDA green fluorescence signal. The latter taking place while nutrients allow for utter activation of photosynthesis, therefore increasing $\alpha$-JV$_{PSII}$, and $\alpha$-ETR. We propose that both light

acclimation and CCMs downregulation may well constitute the underlying mechanism responsible for this cyanobacterium remarkable acclimation capacity and constant growth along the time course of the experiment.

**5 Conclusions**

This study demonstrates that there is little evidence of a detrimental influence of high alkalinity in the natural phytoplankton community that occurs in the oligotrophic environment of the sub-tropical Atlantic Ocean. OAE treatments did not cause negative physiological effects on the community and physiological fitness was not hampered. In general, statistical analyses yielded non-significant results, pointing towards neutral effects of OAE. However, the experimental conditions were clearly differentiated in a principal component analyses (Fig. S10-Supp.Mat) suggesting that the measured variables were effectively

discriminated depending on the OAE treatments. Intermediate OAE data are more dispersed within the PCA space, precisely

explaining the major changes occurring during the experimental treatments. Indeed, the phytoplankton that developed in the intermediate OAE treatments seemed to be able to more effectively recycle and/or utilise nutrients, thus allowing a markedly higher productivity to occur under oligotrophic conditions. The capacity for enhanced net growth under these circumstances was specific to certain groups of phytoplankton. Most of the significant changes in the community took place between t20 and

t33 of the experiment (phase -II). Cell abundances for the nanoeukaryotes > 20μm from the genera *Chrysochromulina*, picoeukaryotes and some of the microplankton significantly increased in this period. In parallel, photosynthetic efficiency was higher and cell viability was not compromised even at the relatively high levels of alkalinity of the intermediate treatments (up to $\Delta1800$ μmol · $L^{-1}$). This did not occur at the lowest and extremely high alkalinity levels.

The mechanism underlying the phytoplankton communities being apparently able to make use of remineralised nutrients to

grow rapidly to high abundances under intermediate OAE conditions is not fully understood, and we do not know at present whether nutrient limitation steered the effects caused by OAE. However, the metaproteome analysis revealed a higher expression and activation of proteins related to N transport, thylakoidal electron transport and the Calvin cycle. Whether this effect of intermediate OAE treatments, enhancing recycling capacity, is a consistent or a repeatable phenomenon in other communities and time frames, remains to be elucidated, as do the precise mechanisms behind it. It may be restricted to

oligotrophic communities under severe nutrient stress. Ecological stress is defined as the impact of any set of abiotic and biotic factors negatively affecting the performance individually, and eventually deteriorating the growth rate of the population through the reduction of individual survival, growth, and reproduction (Grime 2009 Vinebrooke et al., 2004). According to this, we can conclude that OAE did not cause stress in the phytoplankton community studied. Considering the phytoplanktonic compartment of the ecosystem, to evaluate whether the advantages of implementing OAE at a global scale would exceed the

risks or simply, how sustainable OAE is once the treatments are deployed in the sea, other factors primarily associated with seawater chemical alterations must be considered, to safely apply OAE practices in our oceans.

**Data availability**

The raw data supporting the conclusions of this article will be made available by the authors.


**Author contributions**

Experimental concept and design: UR and JA.

Execution of the experiment: All authors.

Data analysis: LR, LJPP, SDA, MS.

Manuscript preparation: LR.

Manuscript revisions: All authors.

**Financial Support**

This study was funded by the OceanNETS project ("Ocean-based Negative Emissions Technologies – analysing the feasibility, risks and co-benefits of ocean-based negative emission technologies for stabilizing the climate", EU Horizon 2020 Research and Innovation Programme Grant Agreement No.: 869357), and the Helmholtz European Partnering project Ocean-CDR ("Ocean-based carbon dioxide removal strategies", Project No.: PIE-0021) with additional support from the AQUACOSM-plus project (EU H2020-INFRAIA Project No.: 871081, "AQUACOSM-plus: Network of Leading European AQUAtic MesoCOSM Facilities Connecting  Rivers, Lakes, Estuaries and Oceans in Europe and beyond").

**Competing interests**

The authors declare that they do not have any competing interests.

**Acknowledgements**

We would like to thank the Oceanic Platform of the Canary Islands (PLOCAN) and its staff for the use of their facilities and for their help with the logistics and organisation of this experiment. We are also grateful to Andrea Ludwig, and Jana Meyer (KOSMOS Logistics and Coordination), to Jan Hennke, Michael Krudewig, and Anton Theileis (KOSMOS technical staff), to Michael Sswat, Carsten Spisla, Daniel Brüggemann, Silvan Goldenberg, Joaquin Ortiz, Nicolás Sánchez, and Philipp Suessle (KOSMOS Scientific Diving and Maintenance Team). We thank Levka Hansen and Kirstin Nachtigall for pigment HPLC analyses. We are grateful to Juan Luis Pinchetti and Antera Martel from the Spanish Algae Bank (Taliarte, Gran Canaria, Spain) for providing us with cultures for FCM calibrations and for letting us use their flow cytometers. We are also in debt to Minerva Espino for daily assistance and willingness during our stay at the experimental site. We thank the reviewer Katherina Petrou and other anonymous reviewer for very insightful and much appreciated comments on the manuscript.

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
