# Peer review of "Ocean Alkalinity Enhancement (OAE) does not cause cellular stress in a phytoplankton community of the sub-tropical Atlantic Ocean"

_EGUsphere, 2024_

## Referee Comment (RC2)

Review for Ramírez et al. (MS No.: egusphere-2024-847)

The paper "OAE does not cause cellular stress in a phytoplankton community of the subtropical Atlantic Ocean" by Ramírez et al. presents valuable and intriguing data on the phytoplankton community, along with the physiological effects of OAE treatment observed during a mesocosm experiment. While the dataset is substantial and noteworthy, the presentation and condensation of the information could be improved to enhance clarity. Restructuring both the results and discussion sections would help emphasize the main message the authors aim to convey.

Main Comments

1) I understand that mesocosm experiments generate extensive data, often published in various sources. However, this paper would benefit from including more comprehensive background information to elucidate the ongoing processes, such as variations in nutrient levels and pH. Specifically, the paper should address how nutrient limitations, which impact cell numbers and cellular fitness, can be disentangled from the effects of increased alkalinity. Providing this context would significantly enhance the reader's ability to grasp the complex interactions at play.

2) I recommend a better differentiation between community data and physiological fitness. A lot of different methods are mentioned (HPLC, Microscopy, Metaproteome data), but not all of the data is shown or discussed later. For example, the authors often refer to size classes (nano-, picoplankton), but the method say that HPLC data as well as microscopic data is available. Why this is not used later to state the different taxonomic groups? See also comment for L 556.

3) Please include more recent publications in the discussion
(see especially this special issue of
Biogeosciences:https://bg.copernicus.org/articles/special_issue1246.html)

**Abstract**

Comment 1
Please indicate in the abstract which OAE treatment was used. This is important as there are other alkalinity enhancement studies using different alkaline materials.

Comment 2
Graphical abstracts should be self-explanatory and abbreviations shown should be known to the research community or mentioned in the abstract, what does "Ci" stand for.

**Introduction**

Comment 3
Overall, the introduction gives a good background to the research field of OAE. However, I miss a better introduction to the pigments measured (section 2.2; VAZ, DD-DT. DD, DT) and it's importance to understand processes possibly impacted by the OAE treatments. What was the hypothesis for the study? What did the authors expect with increased Alkalinity?

**Results**

Comment 4

In the Results section, it is not clear to me why 3.1 "Community composition" and 3.2 "Total phytoplankton community response" are two separate paragraphs. The results section should be more condensed.

**Discussion**

Comment 5
L463: This sentence does not contribute to the understanding of the phytoplankton and physiological data if it is not placed in context or shown in the results/discussion. Either delete this sentence or use the environmental data to explain your data.

Comment 6
L475: the authors state, that "heterotrophic" processes influence the fitness? Can they rule out nutrients?

Comment 7
L493: "Certainly there was nutrient limitation…"When? Is this driving the decline in phase 1? It is hard to follow the statement, if the nutrient data was never shown (see also main comment above)

Comment 8
L501: The DIC data of the experiment is available, why not directly compare with it.

Comment 9
L556 "Phytoplankton flourished during Phase II...", Which groups? Coccolithophores, dinoflagellates, diatoms? Are there winners and losers among these groups during the different alkalinity treatments?

**Minor comments:**

L30 space between "pronounced" and "community"
L51 Please rephrase the sentence. The most likely scenarios are RCP 4.5 or 6. RCP8.5, often referred to as the "worst-case" or "business as usual" scenario, is considered less likely because it assumes very high emissions with no or very minimal climate policies.
L80 Please modify the sentence, it is hard to read, e.g.: "OAE can be achieved through the weathering of alkaline minerals, like limestone or silicate minerals….. olivine.
L91: correct "species", shouldn't be italic font
L92: the last "I" in watsonii should be italic.
L131: sentence structure: "…set of variables presented here…"
L140: please state the real TA values in seawater as well.
L143: Not sure, why here OAE is abbreviated? Not necessary.
L202: sentence"…was firstly was…"
L289: Sentence structure
L299: delete "below"
L311: Parts of this paragraph should be already mentioned in the introduction.
L441: "The ratios DT+DD… can be used as proxies of the cellular physiological status of the cells…". What exactly does a high or low ratio mean? Does a high ration mean a lot of stress? Please explain this in the text and possibly also in the figure and or figure caption.

L502: This is the first time, this abbreviation is introduced, too late given that it is even in the graphical abstract. Please introduce earlier. Suggestion: Use $C_{inorg.}$ Or something similar.
L514: Sentence structure: "Another possible…"
L573: " total change of 20% in the proteome"…a decrease?

**Figure/Tables**

Figure 1: It is hard to compare the data, if all plots have a different scale. In plot E) and F) you could add an axis break. Consider to add a mark for the different phases.

Figure 2a: Same as in Fig. 1, consider to indicate the different phases of the experiment. Move the legend to the right side of the plot.

Figure 2b,c,d,e: Please increase the font size of the x and y axes.

Table 1: Please check the Photophysiology box, it shows two squares. Not sure, if this is intended.

Figure 3a: The labels are a bit blurry which makes it difficult to read. Maybe you can improve the figure.

Figure 4,5: What are the grey and red filled squares? Not explained in the figure caption

Figure 6: The different symbols in the legend below are not necessary, only display the colors of the Alkalinity treatments.

---

## Author Response (AR1)

Response to reviewers' comments on egusphere-2024-847: Ocean Alkalinity Enhancement (OAE) does not cause cellular stress in phytoplankton in a mesocosm experiment by Ramírez et al. 2024

RC1: Katherina Petrou

1.      It would be helpful to have the nutrient information included or summarised in the main text somehow, as the authors for example mention the link between stress, viability and nutrient limitation (especially Phase 1), but there is no visible measure of any nutrients and their dynamics, making assessment of this assertion difficult. While these data have been published elsewhere and are not specifically included here, it could be nice to integrate them visually somehow. Particularly, it would be interesting to know if any data were correlative, i.e. chlorophyll and nutrient dynamics?

Reviewer has made a good point here since nutrients are needed to be able to understand the underlying processes. A brief description of the nutrient dynamics during the experiment has been added as well as the nutrient trends shown in Figure S3-Supplementary material. These plots have been taken from Paul et al, (2024), BG special issue where this will be published. Such an investigation is on to the same mesocosm experiment as ours.

Nitrate, nitrite and phosphate were under the detection limits of the analysis techniques employed (Fig. S3. B, C - supplementary material). Despite that some correlations were significant, looking at Fig. S3 our correlation analysis does not support any interesting finding really. The correlation plots and data distribution among nutrient concentration (Si, $NO_3$, $NO_2$, $PO_4$ µM) and fractioned chlorophyll (dependent variable) on the size ranges just as before Chl-a >20 µm, Chl-a 2-20 µm, Chl-a 0.2-2 µm, and total-Chl-a are shown in Figure S4-Supplementary material though since there was nutrient turnover in phase 2 (most likely organic). Yet, we think this could be eliminated from the final Supplementary material.

The changes made in the results section are as follows: Line 292-301:
*"Phase-0 was established at the beginning of the experiment (t1-t3), representing the conditions in the mesocosms before the treatments were applied. Phase-I (t5-t20) started after the alkalinization gradient with NaHCO3 and Na2CO3. Finally, Phase-II took place from t21 to t33. DIC and TA were stable up to day 21. Indirect abiotic precipitation occurred in the highest treatment ($\Delta 2400$ µmol · $L^{-1}$) lasting until the end of the experiment and leading to slight TA and DIC losses (Fig. S2-supplemental material). All nutrient levels measured were consistently low during the entire experiment (Fig. S3 - supplementary material) and there were no significant differences due to treatments. Silicic acid dropped off 1.5-fold between days 1 and 6 and remained stable around 0.2 µM (average) during the rest of the experimental period (Fig. S3A - supplementary material). Nitrate, nitrite and phosphate were under the detection limits of the analysis techniques employed (Fig. S3B, C - supplementary material). All this data can be consulted for further details in the article by Paul et al. (2024)."*

2.      This trait-based approach lends itself very well to multivariate analyses (PCA, RDA etc) which would allow for community composition to be overlaid with vectors of variables, or a trait-based PCA. I'm aware that time is a co-variable in this experiment, but analyses could perhaps be done through using the identified Phase 1 vs Phase 2 data. There is of course the limitation of n=1 for many of the variables, which I appreciate precludes many analyses, but maybe by using the multiple measurements that form part of phase 1 and phase 2, can provide some level of replication, as the authors have done with the ROS and viability data. Alternatively, could the authors bin treatments to low, moderate and high – using cluster analysis to delineate the treatment groups, based on physiological and pigment data? I just wonder if there's a way to make the patterns clearer and the links more tenable.

A multivariate Principal Component Analysis (PCA) was carried out to gain a deeper understanding of how OAE treatments (state variables) interacted with the biological variables associated with phytoplankton stress (see Fig. S10 in Supplementary Material). The arrows represent the included original variables such as nutrients, pigments, cell viability (FDA), oxidative stress (ROS), and cell abundances. The direction and length of the arrows indicate their contribution and correlation with the principal components. The points or symbols represent the different experimental conditions i.e. OAE treatemnts: Control, Low, Medium and High organized according to the loading factors of the variables. F1 (46.85%) and F2 (22.71%) together explain approximately 69.56% of the total variance, indicating that these two principal components capture most of the variability in the data. Thus, F1 is the component that explains the most variance, followed by F2.

The relationship between variables that PCA has captured are: Control is close to the origin, indicating that these samples are not significantly affected by the variables. The Low conditions cluster in the negative regions of F1 and F2, showing a greater impact of oxidative stress and low cell viability. Pigments (Fuco, DD, DT, Chla) and other variables related to cell abundances (e.g., dinoflagellates, diatoms) are associated with the Medium conditions both positive and negative in F1 and F2. Despite that Medium is more dispersed this precisely explains the major changes occurring during the experiment.The High treatemnt is well-defined in the positive region of F1, where abundances of certain pigments and cell groups predominate.

The experimental conditions are clearly differentiated in the PCA new axis and representation, suggesting that the measured variables are effectively discriminated depending on the treatments (Control, Low, Medium, High).

**3.      None of the FlowCAM data is presented in the main paper, as I understand this is included in another publication. It might be nice however, to integrate these data more strongly to validate the HPLC pigment interpretations, as pigment content can vary based on physiological adjustment as well as taxonomic shifts, often making it hard to infer too much about community structure. For example, the increase in ZEA could be from changes to photoprotection, as stated in the results, or equally an increase in cyanobacteria, which is also mentioned. The authors also mention the increase in peridinin in Phase 1, but don't link that with dinoflagellate abundance (Supp figure). Adding taxonomic data would strengthen this part of the paper.**

We thank the reviewer for the comments. We would like you to be aware that all data for the different taxonomic groups are presented in Figure S6-supplementary material:" *Cell abundance of phyto- and microphytoplankton in cell-ml-1. A) Picoeukaryotes <2 μm; B) Synechococcus spp. < 2 μm C) Nanoeukaryotes 1 2-20 μm; D) Nanoeukaryotes 2 >20 μm; E) Diatoms; F) Dinoflagellates; G) Silicoflagellates; H) Protozoa. Reproduction in with permission of Biogeosciences, Marin -Samper et al. 2024 for A to D".*

However, in this case we will not incorporate the reviewer's suggestion of including this data set in the main text, due to the large amount of information presented in the main manuscript alraedy. We believe that these data should remain in supplemental material. For clarification we suggest consulting Marin-Samper et al, (2024) for further details as well. Nevertheless, some changes have been done to make the text easier to understand.

See as follows L315-320:

 "*The diatoms showed a decrease in cell numbers during all the experiment, except for M9 where they peaked twice in t18 and t23. Dinoflagellates presented a similar trend at the beginning on phase I.  In phase II (t29-31) all the mesocosms showed significant increase in cells numbers in M6 and M3 (Δ1500 and Δ1800, respectively). Silicoflagellates were only detected during phase II…*

A large part of section 4.2 -to which this comment refers to-has been rediscussed in the light of reviewer appreciations. Please see point 5 for further explanations.

**4. Is there a reason only some of the photophysiological data are presented? For example, the authors mention in the methods fitting the RLC to obtain alpha, Ek and Pm, but don't present these results. Equally, authors could look at the photophysiologically derived NPQ values from these data or photoinhibitory parameters, which could be compared with or link to the pigment discussion around xanthophyll ratios (VAZ and DD:DT).**

We agree with the reviewer; the information was incomplete. In addition to Fv/Fm we have now in Fig. 2 panels B-D the phase-averaged values of the Rapid Light Curve derived values with OAE of: B) α-rETR (relative electron transport rate); C) α-JVPSII (volume-specific electron transport rate); D) $E_K$ (light intensity for saturated JVPSII); E) PM (maximum rate of JVPSII); and F) NSV (non-photochemical energy dissipation or normalized Stern-Volmer quenching). The results read now as follows in Lns-343-357:

*" Dark adapted PSII photochemical efficiency (Fv/Fm) of the phytoplankton communities initially exhibited a decline in all mesocosms, and for the remainder of phase-I between t7 and t20, stayed relatively constant, within the range of 0.25 to 0.38 (Fig. 2A). Fv/Fm values were significantly different between levels of alkalinity and between phases, showing a highly significant interactive effect (Table 1). By the beginning of phase-II, Fv/Fm was close to 0.30 in all mesocosms. At the start of phase-II, an increasing trend in Fv/Fm and α-JVPSII (Fig. 2C) was observed in mesocosms with intermediate alkalinity (up to the Δ1800 treatment), with the highest ΔTA mesocosms showing ~2-fold lower values. No significant differences were observed between only alkalinity levels in terms of the photosynthetic parameters α-ETR and α-JVPSII during phase-I or phase-II (Fig. 2B, Table 1).*

*The increasing trend of the initial slope of electron flux per unit volume, α-JVPSII, with increasing alkalinity that paralleled Fv/Fm as it increased with OAE up to ΔTA1800, was not apparent in the statistical analysis for variations in only alkalinity but was apparent as a significant interactive effect of the phases and alkalinity (Table 1), possibly due to the discrepancy in response of the community at ΔTA 1200. In contrast to α-JVPSII, the initial slope from the RLCs of relative PSII electron transport versus PAR, α-ETR, did show differences between phases but not between levels of alkalinity. The maximum rate of JVPSII (PM) and non-photochemical quenching (NSV) did not present significant differences. Yet, NPQ shows an increasing pattern phase II in the intermediate treatments."*

**5. Changes in Fv/Fm can be community structure related. In section starting at L540, authors speculate on the reason for decline in FvFm ratios at the highest TA treatments, but conclude no stress, based on no change in ROS etc. There is little discussion about the influence of how community shift may affect these analyses. The authors conclude no effect of TA, but rather that the intermediate TA mesocosms had relief from nutrient limitation. Again, nutrient data would be helpful here.**

Reviewer has pointed out an important question here: how community shifts affect physiology and in turn, how physiology can affect community development if it is constrained by nutrients. We have made substantial changes in section 4.2 to accommodate all comments raised by the reviewer in points 3, 4 and 5 in a comprehensive manner.

See lns 585-641:

*"4.2 Changes in the community (phase-II)*

*Phase-II occurred between t20 and t33, and several noteworthy changes were observed. Chlorophyll-a (Chl-a) levels increased, in parallel with a significant recovery in the Fv/Fm ratio, and raised α-JVPSII, and α-rP values, particularly in treatments with medium ΔTA (Δ900, Δ1500 and Δ1800 µmol · L-1), apart from Δ1200. Conversely, Fv/Fm values dropped-off in the low and high ΔTA mesocosms during this phase. As mentioned earlier, variations in Fv/Fm can be indicative of stress directly affecting Photosystem II (PSII). Excess irradiance injures phytoplankton by inducing photobleaching of photosynthetic pigments and by altering complexes and molecules present in the photosynthetic apparatus such as the Photosystem II (PSII) complex and/or the rubisco enzyme (Häder et al. 1998) due to overshooting electrons and oxy-radicals (Segovia et al. 2015). However, the cells react to high light conditions by triggering photoprotection mechanisms (Segovia et al. 2015). The most important ones are related to the xanthophyll cycle (oxygenated carotenoids), and other carotenoids regarding non-photochemical quenching (NPQ). The violaxanthin−anteraxanthin−zeaxanthin (VAZ) cycle has been described in chlorophytes, and the diadino- xanthin−diatoxanthin (DD−DT) cycle is present in Bacillariophyceae, Haptophyta, and most of the chl c- containing algae (Takaichi 2001).*

[revised manuscript text omitted]

**Some minor comments:**
**1. L203: 'was firstly' is written twice, please delete one:**
Done

**2. Figure 1: consider shading Phase 1 and 2 on plots – background colour? Or mark with dashed vertical line?**
Done

**3. Line 384: what is meant by 'other' treatments. It is the start of a new paragraph, so perhaps remind the reader which treatments are being discussed.**

We refer to the treatments not mentioned before in: "*Δ1800 and Δ2100 μmol · L⁻¹ experienced 10 % and 12 % significant protein downregulation.*" As the intermediate treatments were only reduced by 10-12%. We have clarified this in the text.

This has been rephrased in L437: *"Protein abundances dropped by approximately half in high and low treatments."*

**4. Line 584: possible spelling error "caseated"?**

Yes, we have changed the sentence by: "*The DD+DT: DT: Chl-a ratio also increased in intermediate ΔTA due to a possible input of nutrients*"

**5. The authors could consider indicating Phase 1 and 2 on fluorescence data plots:**

Done

**6. Similarly, could these data (photophys) be compared statistically? Or possibly by TA grouping Low, Med, High?**

Done

**RC2: Anonymous referee**

**1. I understand that mesocosm experiments generate extensive data, often published in various sources. However, this paper would benefit from including more comprehensive background information to elucidate the ongoing processes, such as variations in nutrient levels and pH. Specifically, the paper should address how nutrient limitations, which impact cell numbers and cellular fitness, can be disentangled from the effects of increased alkalinity. Providing this context would significantly enhance the reader's ability to grasp the complex interactions at play.**

We agree with the reviewer and have added all the suggested data and discussion related to it. Please see responses to specific points 1, 4 and 5 to reviewer RC1 above.

As for PH data, these were not previously included in supplemental material, and we certainly think, as the reviewer, that it is critical to understand the changes in the community. Hence, a new panel on Fig. S2-suppl mat has been added. The figure has been taken from Marin -Samper et al. (2024) with permission from BG.

**2. I recommend a better differentiation between community data and physiological fitness. A lot of different methods are mentioned (HPLC, Microscopy, Metaproteome data), but not all the data is shown or discussed later. For example, the authors often refer to size classes (nano-, picoplankton), but the method say that HPLC data as well as microscopic data is available. Why this is not used later to state the different taxonomic groups? See also comment for L 556.**

We understand reviewer's appreciation. We think that such differentiation is clear on the result sections 3.1 "Community composition" and section 3.2 that reads now: "Phytoplankton physiological response". It is just nor merely the title. As replied to RC1, you may see that in the discussion section, specifically in 3.2, we have made substantial changes, not only in the re-arrangement of paragraphs, but also removing some and added new information that has been obtained after further data analyses, carried out to address all your and RC1 comments. The only data not shown in the main text were corresponding to microplankton, but they are in in Figure S6-supplementary material. HPLC and metaproteome are both in the main text. Additionally, some other complementary metaproteome results were already in supplemental material since the beginning. Most likely there must be a misunderstanding since we have not detected anything of the sort of *"HPLC data as well as microscopic data is available".* HPLC data are available in Figure 1, and microscopy data are available for microplankton in Figure 4H and 5H all the time. We are happy to amend any consideration that the reviewer might still have if this is not satisfactory if we are still missing something.

**3) Please include more recent publications in the discussion (see especially this special issue of Biogeosciences:hSps://bg.copernicus.org/arBcles/special_issue1246.html):**
Done

**4. Abstract**
**Comment 1. Please indicate in the abstract which OAE treatment was used. This is important as there are other alkalinity enhancement studies using different alkaline materials.**

Yes. We have included now in ln 20: *"OAE was based on the addition of carbonates ($NaHCO_3$ and $Na_2CO_3$)"*

**Comment 2. Graphical abstracts should be self-explanatory, and abbreviations shown should be known to the research community or mentioned in the abstract, what does "Ci" stand for.**

We have added a legend for Ci, OAE and $\Delta$TA.

**5.Introduction**
**Comment 3. Overall, the introduction gives a good background to the research field of OAE. However, I miss a better introduction to the pigments measured (section 2.2; VAZ, DD-DT. DD, DT) and it's importance to understand processes possibly impacted by the OAE treatments. What was the hypothesis for the study? What did the authors expect with increased Alkalinity?**

We feel that expanding on pigments in the introduction will interrupt the thread of all the variables that have been measured, losing the general glimpse of the study.

Therefore, we have clarified the pigment questions in the results section in L594-598: "*pigments serve as a proxy because some of them are related to photoinhibition and/or photodamage processes. The xanthophyll cycle (oxygenated carotenoids) is directly related to non-photochemical quenching (NPQ). On the other hand, the violaxanthin-antheraxanthin-zeaxanthin (VAZ) cycle has been described in chlorophytes, and the diadinoxanthin-diatoxanthin (DD-DT) cycle is present in Bacillariophyceae, Haptophyta, and in most algae that contain chlorophyll (Takaichi 2011).*"

Additionally, a thorough discussion on why pigments serve as proxies for cellular stress been included in the discussion section, please see comment 12.

Regarding the second and 3rd questions: The aim and prospective results expected were already mentioned in ln115-123:

"*Based on the previous studies mentioned above, we hypothesized that we would not expect to see potential physiological consequences of increased equilibrated alkalinity, precisely due to equilibration. However, adding carbonate salts to the phytoplankton community could produce transient-quick acclimation processes in phytoplankton cells that most likely will display a reversible-stress response. Such a response might be mediated by nutrients (Falkowski & Raven, 2007). Hence, transcription of genes and the protein inventory will change accordingly. As a result, the metabolic activity (i.e. cell viability), variable chlorophyll fluorescence (related to photosynthetic function), pigments composition and ratios (proxies for photoinhibition and physiological stress) and reactive oxygen species -ROS- accumulation (accounting for oxidative stress management) might also change depending on whether OAE exerts negative, positive, or neutral effects*"

**6.Results**
**Comment 4. In the Results section, it is not clear to me why 3.1 "Community composition" and 3.2 "Total phytoplankton community response" are two separate paragraphs. The results section should be more condensed.**

This has been responded above in point 2 to you and in point 5 to RC1.We hope you both find our responses satisfactory.

**7.Discussion**
**Comment 5. L463: This sentence does not contribute to the understanding of the phytoplankton and physiological data if it is not placed in context or shown in the results/discussion. Either delete this sentence or use the environmental data to explain your data:**

Deleted

**Comment 6.L475: the authors state, that "heterotrophic" processes influence the fitness? Can they rule out nutrients?**

Indeed. As explained before, nutrients have been included now in supplemental material.
In addition, in Ln 531 the following is stated*: "it may also be due to the drop in nutrient concentrations since the beginning of phase I, this depletion in inorganic nutrients may have benefited heterotrophic organisms."*

**Comment 7. L493: "Certainly there was nutrient limitation…" When? Is this driving the decline in phase 1? It is hard to follow the statement, if the nutrient data was never shown (see also main comment above):**

Please see answer to comment 6.

**Comment 8. L501: The DIC data of the experiment is available, why not directly compare with it.**

We think that it is not necessary to compare the DIC here. It is only mentioned as a possible downregulator of CCMs.

**Comment 9. L556 "Phytoplankton flourished during Phase II...", Which groups? Coccolithophores, dinoflagellates, diatoms? Are there winners and losers among these groups during the different alkalinity treatments?**

On Ln 621 we mentioned the group, genera, and pigments relation with the blooming functional groups: "*These significant improvements in Chlorophyll-a (Chl-a), Chlorophyll-c2 (Chl-c2), fucoxanthin and divinyl-Chl-a containing species as well as in photophysiology data, are ascribed to the emergence of blooms of nanoeukaryotes-2 from t23 to t27 when they peaked. Specifically, Haptophytes from the genera Chrysochromulina in Δ1500 and in Δ1800. Picoeukaryotes also peaked in this period in such mesocoms*"

L640: We have included the species and OAE treatments to make it clear in the text: "*Indeed, it is somehow striking that given the high oligotrophy of the system, phytoplankton (Chrysochromulina in Δ1500 and in Δ1800 flourished) flourished during phase-II.*"

We are not sure about refereeing to "winners and losers" in the light of the results obtained, in which not compelling clear effects of OAE came up. True that intermediate TA seemed to be somehow beneficial, but significant differences were not that outstanding. Hence, we have just mentioned those groups that were most likely benefited vs. other that were not.

**Minor comments:**

**1.      L30 space between "pronounced" and "community":**
Done

**2.      L51 Please rephrase the sentence. The most likely scenarios are RCP 4.5 or 6. RCP8.5, often referred to as the "worst-case" or "business as usual" scenario, is considered less likely because it assumes very high emissions with no or very minimal climate policies**

We agree, predictions have changed and so has terminology. We have amended the text including the new models from CMIP6 i.e. the SSP (Shared Socioeconomic Pathway, 6AR IPCC 2021, as reviewer for sure knows).

New text reads as follows, Ln 49-61:

"*We are confronting one of the greatest challenges in the 21st century due to the progressive increase in atmospheric $CO_2$ caused by anthropogenic activities. The 2015 Paris Agreement represented a pivotal moment in global climate policy by setting the objective of keeping global temperature rise well under 2°C. This target was motivated by the urgency to prevent irreversible impacts and mitigate intolerable risks to human society. The Shared Socioeconomic Pathway (SSP) climate simulation-based scenarios (7AR IPCC 2021) combining elements from the new models about future societal development (SSP) with the previous iteration of scenarios, the representative concentration pathways (RPCs), describe new trajectories of changes in atmospheric greenhouse gases (GHG) over time, including very ambitious mitigation policies (Riahi et al, 2017). According to this, the most likely scenarios that we will be facing up to the year 2100 are the "middle of the road" (SSP2-4.5) and the "rocky road" (SSP3-7). The former is an intermediate GHG emission scenario, in which the predicted total cumulative $CO_2$ emissions will reach 600 μatm by 2100. In the latter, high GHG emissions may take place and $CO_2$ concentrations of 900 μatm are to be expected by 2100. This scenario also comprises a concomitant reduction of 0.25 units in the global ocean surface pH by 2100 with respect to 1950 levels for SSP2-4.5 and 0.35 units for SSP3-7, clearly leading to severe ocean acidification (OA).*"

**L80 Please modify the sentence, it is hard to read, e.g.: "OAE can be achieved through the weathering of alkaline minerals, like limestone or silicate minerals….. olivine:**

We have modified the text as suggested in L85-88*:*
"*OAE can be achieved through the weathering of alkaline minerals like, limestone by the addition of carbonates (as hereby presented), silicate minerals such as the iron-containing mineral olivine, through electrochemical processes used to generate alkalinity from chloride brines and by additions of hydrated-carbonate-minerals to seawater and ground minerals to the shelf seafloor.*"

**3.      L91: correct "species", shouldn't be italic font:**

Done

**4.      L92: the last "I" in watsonii should be italic:**
Done

**5.      L131: sentence structure: "…set of variables presented here…":**
Done

*6.*      **L140: please state the real TA values in seawater as well**
Done ",

Changes in L144:  *reaching actual values of total alkalinity (TA) on the seawater between 2300 and 4700 µmol · L-1, respectively."*

**7.      L143: Not sure, why here OAE is abbreviated? Not necessary:**
Done: OAE was deleted.

**8.      L202: sentence "…was firstly was…":**
Corrected

*9.*      **L289: Sentence structure:**

We have rephrased the sentence in L292-295*:*

*"Phase-0 was established at the beginning of the experiment (t1-t3), representing the conditions in the mesocosms before the treatment was applied. Following this, Phase-I (t5-t20) commenced after alkalinization with NaHCO₃ and Na₂CO₃. Finally, Phase-II occurred from t21 to t33. The trends in alkalinity and dissolved inorganic carbon (DIC) are illustrated in Fig. S2 in the supplemental material."*

**10.      L299: delete "below":**
Done

**11.      L311: Parts of this paragraph should be already mentioned in the introduction.**

We are confused about this comment. We do not know exactly to which part you do refer.

*12.*      **L441: "The ratios DT+DD… can be used as proxies of the cellular physiological status of the cells…". What exactly does a high or low ratio mean? Does a high ration mean a lot of stress? Please explain this in the text and possibly also in the figure and or figure caption.**

We concur with the reviewer that this part was not clear enough. We have expanded on pigments functions and explained the meaning of the ratios in Lns 591-618:

*"Excess irradiance injures phytoplankton by inducing photobleaching of photosynthetic pigments and by altering complexes and molecules present in the photosynthetic apparatus such as the Photosystem II (PSII) complex and/or the rubisco enzyme (Häder et al. 1998) due to overshooting electrons and oxy-radicals (Segovia et al. 2015). However, the cells react to high light conditions by triggering photoprotection mechanisms. The most important ones are related to the xanthophyll cycle (oxygenated carotenoids), and other carotenoids regarding non-photochemical quenching (NPQ). The violaxanthin−anteraxanthin−zeaxanthin (VAZ) cycle has been described in chlorophytes, and the diadino- xanthin−diatoxanthin (DD−DT) cycle is present in Bacillariophyceae, Haptophyta, and most of the chl c- containing algae (Takaichi 2001).*

*Our  data suggests that under moderate ∆TA conditions, the community experienced a relaxation in its photosynthetic state since Fv/Fm increased, indicating the increasingly effective transfer of electrons to photochemistry, without causing the accumulation of reactive oxygen species (ROS). Such relaxation is further supported by the lack of an increase in accessory pigments (Fig. 1) and the behaviour of the DD+DT: TFUCO and DD+DT: Chl-a (mol: mol) ratios (Fig. 6). In multi-species bloom situations, the*

*ratios of DT+ DD:TFUCO and DT+DD:chl-a can be used as an indicator of the cellular physiological status (Stolte et al. 2000). Indeed, what in part determines phytoplankton species fitness, succession and distribution are the physiological trade-offs between light harvesting, photoacclimation or photoprotection and dissipation of excess energy, including oxy-radicals, in which all pigments participate. This is reflected by changes in the pigment ratios of DT+DD:TFUCO and HEXA:TFUCO indicating physiological stress (Stolte et al. 2000). Upon examining DD+DT: TFUCO, we observed a significant increase of this ratio in intermediate ΔTA. Fucoxanthins serve as a highly efficient light-harvesting pigment when irradiance is optimal (Harris et al. 2009). The lower these ratios are, the more stress is exerted upon the cells. For instance, a lowered (DT+DD): TFUCO (mol: mol) ratio implies a dilution of the DD−DT cycle, meaning that the cells' capacity for non-photochemical quenching (NPQ) is decreased, being prone to excessive ROS and photodamage (Segovia et. al 2018). NPQ increased towards the middle of phase II coincidentally with bloom peaks in the intermediate treatments (Fig. S7-supplemental material). In parallel, increases in DD+DT:TFUCO were observed . A higher NPQ means that the capacity of the cells to dissipate excess energy is enhanced, thus we can exclude photodamage as a direct responsible for Fv/Fm declines. Additionally, no differences were observed between treatments suggesting that the OAE treatments did not negatively influence the photophysiological state of the phytoplankton community. It is important to note that all these pigments are involved in the long-term photoresponse (Dimier et al. 2009), explaining their absence during phase-I. Along this line, the DD+DT: Chl-a ratio also might as well have increased in intermediate ΔTA due to a possible input of nutrients."*

**Figure/Tables**

**1.      Figure 1: It is hard to compare the data, if all plots have a different scale. In plot E) and F) you could add an axis break. Consider to add a mark for the different phases.**

We had to do it this way because if all the graphs have the same ranges on the y-axis, we cannot see the dynamics of all the data represented clearly, due to the differences in concentrations between the different pigments. On the other hand, we have added the marks to differentiate between phases.

**2.      Figure 2a: Same as in Fig. 1, consider to indicate the different phases of the experiment. Move the legend to the right side of the plot: Done**

We indicated the different phases of the experiment with a discontinuous red line in t21, when phase II starts.

**3.      Figure 2b,c,d,e: Please increase the font size of the x and y axes.:**
Done

**4.      Table 1: Please check the Photophysiology box, it shows two squares. Not sure, if this is intended:**
Done

**5.      Figure 3a: The labels are a bit blurry which makes it difficult to read. Maybe you can improve the figure.**

**6.      Figure 4,5: What are the grey and red filled squares? Not explained in the figure caption**
We explain in the figure caption that these graphs are *whisker-plots* representing the percentage of viable cells. We add the clarification on the grey and salmon colors for the different phases.

See L467 and L470:
*"Figure 4. Box and whisker-plots of the percentage of viable cells, determined using the FDA green fluorescence stain for phases-I and II respectively, for all functional groups. Grey color for phase I and salmon color for phase II. A, B) Synechococcus < 2 μm; C, D) picoeukaryotes < 2 μm; E, F) nanophytoplankton-1, <20μm; G) nanophytoplankton -2, > 20μm; H) microphytoplankton (30-280 μm).*

**7.      Figure 6: The different symbols in the legend below are not necessary, only display the colors of the Alkalinity treatments:**

We did include the symbols for color blind people. We believe they should remain.

We thank the reviwers for insighful comments and we hope that they find our responses satisfactory

Sincerely,

Librada Ramírez

---

## Author Response (AR2)

**Response to reviewers' comments on egusphere-2024-847- R2:** Ocean Alkalinity Enhancement (OAE) does not cause cellular stress in phytoplankton in a mesocosm experiment by Ramírez et al. 2024

**RC2: Anonymous referee**

**I have read the response and the revised manuscript. The authors have done an excellent job addressing all the reviewers' comments. However, I have a few minor comments, mainly regarding the abstract and the conclusion, to give a precise and concise study result. This primarily concerns the wording and requires only a few adjustments.**

**1.-In the Abstract it is stated "Taken all data together, this study shows that there is minimal evidence indicating a harmful impact of high alkalinity on the plankton community.**

**The Conclusions says" According to this, we can conclude that OAE did not cause stress in the phytoplankton community studied".**

**Result: "none of the groups showed significant changes with respect to treatments except for Synechoccocus ssp. (Fig. 4B; Table 1).**

**Is there "little evidence" or not. I think the author should be more precise. Please modify the abstract and conclusions accordingly and also make clear that most of the drawdown in phytoplankton was caused by nutrient limitations (N and P below detection limit), not by Alkalinity increase.**

Agree. Seems we did not provide a cristal clear message afterall. We have changed the last part of the abstract. It now reads in Lns 39-41 as :

 *"Taken all data together, this study shows that OAE did not cause cellular stress in the phytoplankton community studied and physiological fitness was not impaired. Most likely the drawdown in phytoplankton cell numbers observed at times seemed to have been caused by nutrient limitation. "*

And in ln 726-727as:

*"This study demonstrates that there is no evidence of a detrimental influence of high alkalinity in the natural phytoplankton community that occurs in the oligotrophic environment of the sub-tropical Atlantic Ocean…"*

**2.-L3: Rather add the information, how many mesocosms instead of the volume (or both).**

Done

**3.-L144: "Δ0 (lowest) -Δ2400 (highest), better use "to" or "and" In between both alkalinity levels.**

Done

**4.-Figure 1: Maybe you could mark the start of the OAE treatment as well with a dashed line**

Done

**5.-Figure 6: The numbers for the different Alkalinity treatments are very small. It would be good to have an extra legend in the figure, as none of the other figures in the main text include the gradient as a legend.**

Done. We have also inlcuded the legend in Fig 2A.

We thank the reviwer for more helpful comments that help to the final polishing of the ms. and hope that our responses are found satisfactory.

Sincerely,

María Segovia